# Interpretable Traces, Unexpected Outcomes: Investigating the Disconnect in Trace-Based Knowledge Distillation

## Abstract

Recent advances in reasoning-oriented Large Language Models (LLMs) have been driven by the introduction of Chain-of-Thought (CoT) traces, where models generate intermediate reasoning traces before producing an answer. These traces, as in DeepSeek R1, are not only used to guide model inference but also serve as supervision signals for Knowledge Distillation (KD) to improve smaller models. A prevailing but under-examined implicit assumption is that these CoT traces are both semantically correct and interpretable for the end-users. While there are reasons to believe that these intermediate tokens help improve solution accuracy, in this work, we question their validity (semantic correctness) and interpretability to the end user. To isolate the effect of trace semantics, we design experiments in the Question Answering (QA) domain using a rule-based problem decomposition method. This enables us to create Supervised Fine-Tuning (SFT) datasets for LLMs where - each QA problem is paired with either verifiably correct or incorrect CoT traces, while always providing the correct final solution. Trace correctness is then evaluated by checking the accuracy of every sub-step in decomposed reasoning chains. To assess end-user trace end-user interpretability, we also finetune LLMs with three additional types of CoT traces: DeepSeek R1 traces, LLM-generated summaries of R1 traces, and LLM-generated post-hoc explanations of R1 traces. We further conduct a human-subject study with 100 participants asking them to rate the end-user interpretability of each trace type on a standardized Likert scale. Our experiments reveal two key findings - (1) Correctness of CoT traces is not reliably correlated with the model's generation of correct final answers: correct traces led to correct solutions only for 28% test-set problems while incorrect traces don't necessarily degrade solution accuracy. (2) In end-user interpretability studies, fine-tuning on verbose DeepSeek R1 traces produced the best model performance but these traces were rated as least interpretable by users, scoring on average 3.39 for end-user interpretability and 4.59 for cognitive load metrics on a 5-point Likert scale. In contrast, the decomposed traces that are judged significantly more interpretable don't lead to comparable solution accuracy. Together, these findings challenge the assumption in question suggesting that researchers and practitioners should decouple model supervision objectives from end-user-facing trace design.

## 1 Introduction

Reasoning with intermediate Chain-of-Thought (CoT)-style traces (step-by-step outputs that models produce prior to an answer) has become one of the defining strategies for improving the performance of Large Language Models (LLMs) over a diverse range of problems, as exemplified by approaches like DeepSeek R1 (Guo et al., 2025). While models such as DeepSeek R1 often produce extremely verbose unstructured responses even for simple problems (Kambhampati et al., 2025), these reasoning traces are utilized both as inference aids and supervision signals in Knowledge Distillation (KD) when Supervised Fine-Tuning (SFT) smaller LLMs for enhanced task performance (Magister et al., 2022; Shridhar et al., 2022; Tian et al., 2025).

A common but often implicit assumption behind these CoT traces is that they are semantically correct and interpretable for end-users (Guo et al., 2025). Training with these traces is done primarily to improve LLM performance on a given task, but training/fine-tuning objectives rarely require these traces to be semantically

correct or interpretable [1]. In this work, we challenge this assumption and ask: "*Must CoT reasoning traces be semantically correct and interpretable to end-user for enhancing LLM task performance?*"

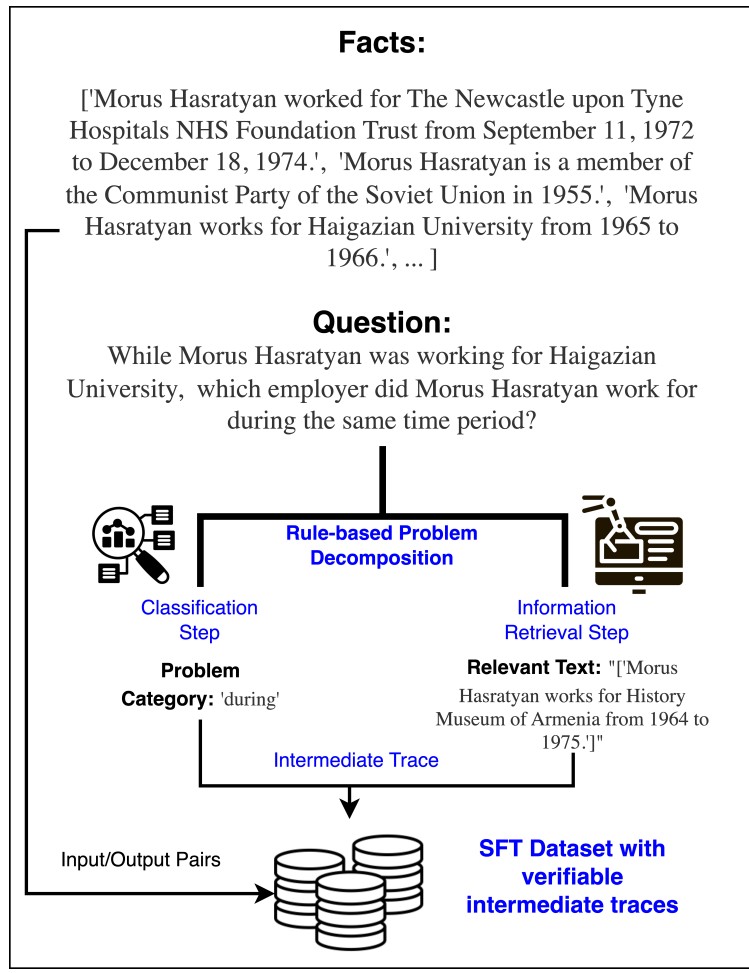

<div>

**Facts:**

['Morus Hasratyan worked for The Newcastle upon Tyne Hospitals NHS Foundation Trust from September 11, 1972 to December 18, 1974.', 'Morus Hasratyan is a member of the Communist Party of the Soviet Union in 1955.', 'Morus Hasratyan works for Haigazian University from 1965 to 1966.', ... ]

**Question:**
While Morus Hasratyan was working for Haigazian University, which employer did Morus Hasratyan work for during the same time period?

**Rule-based Problem Decomposition**

Classification Step          Information Retrieval Step

**Problem Category:** 'during'     **Relevant Text:** "['Morus Hasratyan works for History Museum of Armenia from 1964 to 1975.']"

Intermediate Trace

Input/Output Pairs          **SFT Dataset with verifiable intermediate traces**

</div>

Figure 1: The construction of SFT dataset w/ verifiable intermediate traces using rule-based problem decomposition on an example from the CoTemp QA dataset. Rule-based decomposition allows us to algorithmically generate the steps for the intermediate trace for the training data, also allowing us to verify these similarly structured traces produced by the model during inference.

To address this, we focus our experiments on the Question Answering (QA) domain, where end-users regularly interact with both intermediate traces and final outputs (Polemi et al., 2024) (e.g., ChatGPT (OpenAI, 2023), Perplexity (AI, 2023), Copilot (Microsoft, 2023), Gemini (Google, 2023)). Faithfulness of reasoning traces is especially critical in these interactive settings where unverifiable traces can lead to loss of trust in users, misinformation and errors in model outputs, and perpetuation of biases among other negative consequences (Guidotti et al., 2018). To assess the trade-offs between semantic correctness of the traces and LLM performance, we design an experimental setting where both final solutions and intermediate traces can be independently evaluated. Specifically, we employ a rule-based problem decomposition technique to break QA tasks into structured sub-problems (McDonald & Emami, 2024; Xue et al., 2024). Next, we generate SFT datasets pairing questions with either verifiably correct or verifiably incorrect reasoning traces (while always including the correct answer). At inference, this allows us to verify the correctness of both the final solution and the intermediate traces generated by the distilled model. Note, that "validity" and "semantic correctness" of the intermediate traces in this context refer specifically to the alignment with the rule-based decomposition (Category and IR steps as shown in Figure 1).

To assess the trade-offs between end-user interpretability and LLM performance, we fine-tune models on different types of reasoning traces: DeepSeek R1 traces (verbose CoT outputs), LLM (GPT-4o-mini)-generated summaries of R1 traces (end-user facing summarizations), LLM (GPT-4o-mini)-generated post-hoc explanations (natural language explanations of R1 traces), and verifiably correct traces that we discussed above. In parallel, we conduct a human-subject study with 100 participants (hired on Prolific), split into four sets of 25. Each group was asked to judge the interpretability of the trace types using a Likert Scale measuring predictability, comprehensibility, and faithfulness attributes (Jalali et al., 2023; Doshi-Velez & Kim, 2017).

Our experiments reveal two key findings:

---

[1]This distinction has also been brought to light by the recent GPT-OSS models split the trace into three parts - CoT trace, summary and the final answer (OpenAI, 2025). The assumption is that this suggests that GPT-5 is only showing the summary to the users.

1. Correctness of CoT traces is not reliably correlated with LLMs producing correct final answers: correct traces led to correct solutions only for 28% test-set problems, while incorrect traces did not consistently degrade answer accuracy.

2. End-user interpretability of CoT traces is not reliably correlated with LLMs producing correct final answers: fine-tuning on verbose DeepSeek R1 traces led to the strongest task performance, yet users rated these traces as least interpretable, scoring on average 3.39 for end-user interpretability and 4.59 for cognitive load metrics on a 5-point Likert scale.

These results highlight that **semantic correctness and human interpretability of reasoning traces can in fact be an albatross from the perspective of LLM's task performance**, challenging assumptions in current LLM supervision practices.

The rest of the paper is structured as follows: We discuss the relevant existing literature on Large Language & Reasoning Models, Knowledge Distillation, and CoT Trace Interpretability in §2. In §3, we discuss the problem setup and the rule-based problem decomposition for the Open Book QA setting, and how we construct the SFT dataset for distilling the LLMs with correct and incorrect intermediate traces paired with correct solutions. We discuss our experimental setup for SFT experiments and the human-subject studies in §4, followed by an analysis of the results and discussion on the key takeaways from our work in §5. Finally, we conclude our work in Section 6, followed by a note on the Broader Impact of this study. Supplementary includes additional experiment details, and SFT datasets and code will be released on acceptance.

## 2 Related Work

### 2.1 Large Reasoning Models & Chain-of-Thought traces

Large Language Models (LLMs) have shown remarkable performance on a wide variety of natural language tasks in question answering, text generation, summarization, and translation, to name a few (Bubeck et al., 2023). Recent advances in post-training techniques have led to the rise of Large Reasoning Models (LRMs) such as DeepSeek R1 (Guo et al., 2025), Google Gemini 2.5 (Google, 2023), Microsoft Phi-4-reasoning (Abdin et al., 2025), etc. These reasoning models produce a set of intermediate tokens, commonly referred to as 'reasoning' traces, followed by the final solution. While LRMs have shown a significant improvement in final solution accuracy on reasoning tasks over standard LLMs (Guo et al., 2025; Abdin et al., 2025), their intermediate traces are subjective and verbose (Wei et al., 2022; Wang et al., 2022), making it hard to evaluate their trace validity and end-user interpretability (Kambhampati et al., 2025).

### 2.2 Knowledge Distillation and Structured Reasoning Traces

While Small Language Models (SLMs) offer a computationally efficient alternative to LLMs and LRMs, they are not robust to prompt augmentations (such as Chain-of-Thought) or steerable using in-context examples used in few-shot prompt settings (Shridhar et al., 2022; Stolfo et al., 2022). Knowledge Distillation is a well-studied approach used for fine-tuning these SLMs (student) via the outputs of a larger model (teacher) (Magister et al., 2022). With LRMs generating both an intermediate trace and the final solution, SLMs are also distilled to replicate this output (Shridhar et al., 2022; Tian et al., 2025). However, the lack of structured intermediate trace outputs makes the validity of the traces hard to evaluate (Zhou et al., 2022; Chen et al., 2022). This problem is exacerbated for end-user settings such as in Question Answering (QA) domains, where user interactions involve exposure to both intermediate traces and final outputs.

### 2.3 Interpretability of Reasoning Traces

Some recent works have argued for making these CoT traces more interpretable to the end-user, i.e., improve their faithfulness for the end user, as they are believed to serve as the LLM's explanation to generate the final solution (Arcuschin et al., 2025; Tanneru et al., 2024; Li et al., 2024; Tutek et al., 2025; Paul et al., 2024; Lyu et al., 2023; Lanham et al., 2023; Yeo et al., 2024). On the other hand, there has also been work showcasing why these traces are not explainable to the end user (Barez et al., 2025). Both sides of this argument stem

from the assumption that these traces are indeed meant to be useful and interpretable for the end user and not just for the LLM to improve its final solution performance over a certain task. We specifically challenge this assumption and show the disconnect between the use of CoT traces for the LLM (as a training signal in SFT) and the use of CoT traces for the end user (as an interpretable reason behind the model's final solution).

## 3    Knowledge Distillation using Problem Decomposition

In this section, we discuss the rule-based problem decomposition technique that we employ to break down complex Open Book QA reasoning problems into independently verifiable sub-problems (§3.1). We further discuss how we utilize this approach to construct the structured intermediate traces for distilling SLMs (§3.2).

### 3.1    Rule-based Problem Decomposition

In the context of Open Book QA, consider the example shown in Figure 1 which consists of a text passage (referred to as set of facts for our discussion) and a question involving temporal reasoning between the queried problem and the facts present in the provided text. Answering this reasoning question involves identifying the relevant fact from the text which satisfies the temporal relation asked in the problem. In this case, the queried fact refers to *"Morus Hasratyan works for Haigazian University from 1965 to 1966."* The temporal relation queried in the problem is 'during' and thus, the relevant fact that answers the query is *"Morus Hasratyan works for History Museum of Armenia from 1964 to 1975."* Hence, the final answer is *'History Museum of Armenia'.* From this example, we see that the complex Open Book QA problem can be decomposed into a 1) Classification step determining the type of question asked ('during' temporal relation in this case), and an 2) Information Retrieval (IR) step to determine the relevant part of text that can answer the query (the fact with the temporal overlap with the one in question). Therefore, we utilize these two steps to decompose the Open Book QA problems that allow us to construct structured intermediate traces for evaluation.

### 3.2    Intermediate Trace Generation for SFT

Given the outputs of the sub-problems obtained by decomposing the original query as shown in Figure 1, we generate the intermediate traces in an automated way which consists of the Classification step describing the type of the question posed in the query, and the IR step showing the relevant fact in the text that can help answer the query. We construct a dataset using these Input-Trace-Output tuples that can be utilized to SFT the Small Language Models. Note, that by constructing the intermediate trace using these two steps, we can then evaluate the accuracy of the intermediate traces generated by the distilled model at the time of inference. We will refer to this setting as **SFT w/ Correct Traces** for further discussion.

To critically understand the correlation between intermediate trace correctness and final solution accuracy for Knowledge Distillation methods, we also consider an alternative SFT setting where for every input problem, we choose an incorrect problem category and incorrect fact/s for constructing the intermediate trace. This allows us to construct a SFT dataset which also consists of Input-Trace-Output tuples but with incorrect traces and correct final outputs. We will refer to this setting as **SFT w/ Incorrect Traces**. We discuss the empirical setup for our experiments in the following section.

## 4    Experimental Setup

In this section, we first introduce the Open Book QA datasets used for our experiments (§4.1). Next, we discuss the experimental details to assess the correlation between semantic correctness of traces and LLM's task performance (§4.2). Finally, we talk about the SFT experiments and human-subject study that we conduct to assess the end-user interpretability vs LLM performance trade-offs (§4.3). Implementation details for running all our experiments are provided in §4.4.

### 4.1 Datasets

We run our experiments on the following three publicly available Open Book QA datasets:

**CoTemp QA:** CoTemp QA (Su et al., 2024) consists of English co-temporal questions which involve identifying the type of temporal relation posed in the problem, followed by inferring which fact in the given passage of text satisfies the temporal relation with the question. The dataset is categorized into four temporal relation types, namely - 'equal', 'overlap', 'during' and 'mix', and requires around one or two facts for answering the question. For our experiments, we utilize 3,798 train and 950 test samples to construct the SFT datasets.

**Microsoft MARCO QA:** The Microsoft MAchine Reading COmprehension (MARCO) dataset (Bajaj et al., 2016) is an English dataset that consists of a real user-generated queries collected on the Bing platform. Among the other datasets that we use for our experiments, Microsoft MARCO provides the largest passage for each user query generated by a list of URLs in support of answering the question. There are five categories in the dataset, namely - 'description', 'numeric', 'entity', 'location', and 'person', and also requires one paragraph from the passage to answer the question. For our experiments, we utilize 5,000 samples for the train and 1000 samples for test dataset.

**Facebook bAbI QA:** The Facebook bAbI QA dataset (Weston et al., 2015) is also an English dataset that evaluates reading comprehension via question answering problems which requires different reasoning approaches to solve the queried problem such as chaining multiple facts or using deduction. There are 20 question categories in the original dataset, however we utilize 11 categories for our experiments, namely - 'single-supporting-fact', 'two-supporting-facts', 'two-arg-relations', 'counting', 'lists-sets', 'conjunction', 'time-reasoning', 'basic-deduction', 'basic-induction', 'positional-reasoning', and 'size-reasoning'. Each question requires on average three facts to answer the question. We construct the SFT dataset consisting of 3,773 train and 376 test samples. The train/test splits for all the datasets can be found in §A.1.

### 4.2 Trace Correctness vs LLM Task Performance

For our experiments to evaluate the correlation between intermediate trace correctness and final solution accuracy, we utilize the Llama-3.2-1B-Instruct and the Qwen3-1.7B chat models. We adopt the following baselines for our evaluations:

1. **Direct Prompting SLMs:** We directly prompt the two SLMs to establish the baseline performance of these models across the three datasets without any additional fine-tuning.

2. **SFT - Vanilla:** Following the conventional fine-tuning technique, we also utilize the SFT baseline where we fine-tune the models using only Input-Output pairs and no intermediate traces. This allows us to evaluate the final solution performance for these models against the final solution performance obtained via directly prompting and via SFT with intermediate traces.

For examining the impact and correctness of intermediate traces, we run the following experiments:

1. **SFT w/ Correct Traces:** Using the intermediate traces constructed via problem decomposition (Section 3.2), we fine-tune the models using the Input-Trace-Output tuples for each of the three datasets.

2. **SFT w/ Incorrect Traces:** In this case, we construct incorrect intermediate traces as discussed in Section 3.2, but use the correct final solutions in the Input-Trace-Output tuples.

For our experiments on SFT w/ Correct Traces and SFT w/ Incorrect Traces, we additionally report Category Accuracy (signifying the Classification step performance in the intermediate trace), the IR step Accuracy (signifying the IR step performance in the intermediate trace), and average trace length (# of words in the intermediate trace) on the test datasets computed at inference time.

Table 1: An example from the CoTemp QA dataset showing the outputs of Qwen3-1.7B and Llama-3.2-1B-Instruct models under different query setting. Correct final solutions are shown in green, and incorrect final solutions are shown in red. Correct intermediate traces are shown in blue, and incorrect intermediate traces are shown in red.

| Model | Query Setting | Example |
|---|---|---|
| | | **Input Prompt:** 'Answer the question based on the context: [\'Morus Hasratyan worked for The Newcastle upon Tyne Hospitals NHS Foundation Trust from September 11, 1972 to December 18, 1974.\', \'Morus Hasratyan is a member of the Communist Party of the Soviet Union in 1955.\', \'Morus Hasratyan works for Haigazian University from 1965 to 1966.\', "Morus Hasratyan worked for Bishop\'s University from 1972 to 1975."', \'Morus Hasratyan worked for ISCTE – Lisbon University Institute from June, 1957 to December, 1960.\', \'Morus Hasratyan works for History Museum of Armenia from 1964 to 1975.\', \'Morus Hasratyan worked for Royal Air Force College Cranwell in February, 1959.\', \'Morus Hasratyan worked for University of Detroit Mercy in September, 1963.\', \'Morus Hasratyan worked for Tagesspiegel from May, 1957 to November, 1957.\', \'Morus Hasratyan worked for North Carolina State University in May, 1962.\'] Question: While Morus Hasratyan was working for Haigazian University, which employer did Morus Hasratyan work for during the same time period? Only return the answer. |
| | | **Gold Answer:** "['History Museum of Armenia']" ; **Problem Category:** 'during' ; **Relevant Text:** "['Morus Hasratyan works for History Museum of Armenia from 1964 to 1975.']" |
| **Qwen3-1.7b** | Prompt | "Bishop's University" |
| | SFT - Vanilla | "['History Museum of Armenia']" |
| | SFT w/ Correct Traces | "<think>The temporal relation between the event in question and the event in context is: equal. I need to use the following facts to answer the question: ['Morus Hasratyan works for History Museum of Armenia from 1964 to 1975.']</think> <answer>['History Museum of Armenia']</answer>" |
| | SFT w/ Incorrect Traces | "<think>The temporal relation between the event in question and the event in context is: equal. I need to use the following facts to answer the question: Morus Hasratyan worked for ISCTE - Lisbon University Institute from June, 1957 to December, 1960.</think> <answer>['History Museum of Armenia']</answer>" |
| **Llama-3.2-1B-It** | Prompt | "Bishop's University" |
| | SFT - Vanilla | "['History Museum of Armenia']" |
| | SFT w/ Correct Traces | "<think>The temporal relation between the event in question and the event in context is: during. I need to use the following facts to answer the question: ['Morus Hasratyan works for History Museum of Armenia from 1964 to 1975.']</think> <answer>['History Museum of Armenia']</answer>" |
| | SFT w/ Incorrect Traces | "<think>The temporal relation between the event in question and the event in context is: overlap. I need to use the following facts to answer the question: Morus Hasratyan worked for Royal Air Force College Cranwell in February, 1959.</think> <answer>['History Museum of Armenia']</answer>" |

## 4.3 Interpretability of Traces vs Final Solution Accuracy

For our experiments to evaluate the correlation between end-user interpretability of intermediate traces and final solution accuracy, we also include evaluations using larger models (Qwen3-8B and Llama-3.1-8B) to further examine the impact of SFT with more complex traces. Owing to the practical constraints of conducting human-subject studies, we run our experiments on the CoTemp QA domain (§4.1).

### 4.3.1 Reasoning Trace Generation

We consider (1) DeepSeek R1 traces where we prompt the R1 model on the CoTemp QA training dataset and collect the model responses for our SFT experiments where it got the correct final answer. Utilizing this filtered training dataset, we prompt GPT-4o-mini to generate both (2) summaries and (3) post-hoc explanations of these R1 traces. Since R1 traces can often be verbose, we posit that their summary as well as a post-hoc explanation can likely be more interpretable to the end user (prompts shown in §A.2).

### 4.3.2 Human-Subject Study

We conducted four separate user studies to evaluate the end-user interpretability of the four types of reasoning traces. In each study, a set of 25 participants were hired on Prolific and shown only one type of trace: (1) DeepSeek R1 traces, (2) summarized R1 traces, (3) post-hoc explanations of R1 traces, or (4) verifiably correct

reasoning traces. An example of the four types of traces can be found in §B.5. We use a between-subjects design to avoid bias from having participants compare multiple trace types themselves. We specifically test the following hypotheses:

1. **H1:** Reasoning traces that improve task accuracy will not lead to higher interpretability for the user.

2. **H2:** Reasoning traces that improve task accuracy will be associated with higher cognitive workload for the user, as measured by increased mental demand, effort, and frustration.

Each participant viewed five Q/A examples (fixed across all studies), consisting of the input question, the predicted answer, and the reasoning trace. After each example, participants rated the reasoning trace on a 5-point Likert scale on the following properties as suggested by (Doshi-Velez & Kim, 2017; Jalali et al., 2023): predictability, comprehensibility, interpretability, and faithfulness to context (alignment with given facts). To capture the cognitive workload involved in processing and evaluating the traces, we used the NASA–TLX assessment (Hart, 2006), focusing on the dimensions of mental demand, effort, and frustration. Additional user study details, human participant demographics, and study procedure have been discussed in §B.

## 4.4 Implementation Details

Models were fine-tuned using the Hugging Face library (Wolf et al., 2020) on a single 80GB NVIDIA Tesla A100 GPU for 3 epochs (effective batch size 16, max sequence length 1024). We employed PEFT QLoRA (Dettmers et al., 2023) (rank 16, alpha 32) with a learning rate of 2e-4 (8-bit AdamW, cosine scheduler, 0.1 warm-up). Prompt experiments utilized vLLM (Kwon et al., 2023).

## 5 Results

We first discuss the Final Solution Accuracy comparing all four query settings (Tables 2 and 3). Next, we discuss the Intermediate Trace Accuracy comparing the two methods of SFT with correct and incorrect intermediate traces. Finally, we analyze the correlation, or as our results reveal the lack thereof, between Final Solution Accuracy and Intermediate Trace Accuracy (Figures 2 and 3).

Table 2: CoTemp QA Results

| Model | Query Setting | Final Solution Evaluations | | | | Intermediate Trace Evaluations | | |
| | | Accuracy | F1 | Precision | Recall | Classification Step Accuracy | IR Step Accuracy | Avg Trace Length (# tokens) |
|---|---|---|---|---|---|---|---|---|
| Qwen3-1.7b | Prompt | 6.35 | 11.35 | 14.33 | 10.1 | - | - | - |
| | SFT - Vanilla | 60.33 | 74.88 | 82.15 | 71.3 | - | - | - |
| | SFT - Correct Trace | 52.88 | 70.63 | 79.45 | 66.33 | **47.06** | **78.99** | 45.8 |
| | SFT - Incorrect Trace | **63.88** | **76.5** | **82.58** | **73.5** | 20.36 | 56.92 | 34.15 |
| Llama-3.2-1B-It | Prompt | 7.48 | 13.78 | 17.58 | 12.15 | - | - | - |
| | SFT - Vanilla | 44.65 | 61.08 | 69.53 | 56.58 | - | - | - |
| | SFT - Correct Trace | 39.55 | 56.83 | 65.83 | 52.5 | **39.09** | **79.4** | 43.51 |
| | SFT - Incorrect Trace | **45.58** | **61.15** | **69.65** | **57.23** | 18.8 | 73.62 | 40.28 |

Table 3: Microsoft MARCO QA and Facebook bAbI QA Results

| Model | Query Setting | Microsoft MARCO QA | | | | Facebook bAbI QA | | | |
| | | Avg Final Solution Accuracy (%) | Avg Trace Acc (Classification Step) (%) | Avg Trace Acc (IR Step) (%) | Avg Trace Length (# tokens) | Avg Final Solution Accuracy (%) | Avg Trace Acc (Classification Step) (%) | Avg Trace Acc (IR Step) (%) | Avg Trace Length (# tokens) |
|---|---|---|---|---|---|---|---|---|---|
| Qwen3-1.7B | Prompt | 0 | - | - | - | 0 | - | - | - |
| | SFT - Vanilla | 3.4 | - | - | - | **97.9** | - | - | - |
| | SFT - Correct Trace | **26.3** | **60.4** | 40.6 | 68.14 | 94.41 | **60.64** | **24.73** | 43.25 |
| | SFT - Incorrect Trace | 20.3 | 6.9 | **52.5** | 85.07 | 95.21 | 17.82 | 0 | 42.45 |
| Llama-3.2-1B-It | Prompt | 1.7 | - | - | - | 12.8 | - | - | - |
| | SFT - Vanilla | 33.4 | - | - | - | **96.5** | - | - | - |
| | SFT - Correct Trace | **33.7** | **59.9** | 21.4 | 55.82 | 94.41 | **61.7** | **24.73** | 42.17 |
| | SFT - Incorrect Trace | 28.9 | 20 | **43.9** | 80.48 | 86.17 | 3.46 | 0 | 38.5 |

### 5.1 Final Solution & Intermediate Trace Performance

From Table 2, we observe that across all four metrics of Accuracy, Precision, F1, and Recall, SFT w/ Incorrect Trace performs the best in Final Solution Evaluations for both language models. However, by virtue of making the intermediate traces verifiable, we observe that the models which were SFT-ed w/ Correct Traces naturally have higher Classification Step and IR Step accuracies than the models SFT-ed w/ Incorrect Traces. Similar results are observed for the MARCO QA and bAbI QA datasets in Table 3, where both models SFT-ed with intermediate traces have comparable performance in Final Solution Accuracy, but the models SFT-ed with correct traces have better intermediate trace evaluations. These results are pronounced for the bAbI QA dataset (consists of 11 categories for the Classification Step) where the two models SFT-ed w/ Incorrect Traces have negligible intermediate trace performance.

In the case of incorrect traces helping the model improve performance, we hypothesize that this boost in performance can possibly be attributed to the traces' structural patterns seen by the model during training ignoring the semantic aspects. Both, correct and incorrect traces share the same structure in our controlled study setup, and the SFT learning regime that utilizes cross entropy loss possibly enforces the model to learn and output the <incorrect trace, correct final answer> at inference time.

### 5.2 (Lack of) Correlation b/w Final Solution Accuracy & Trace Correctness

From the confusion matrices for SFT models w/ Correct Traces shown in Figure 2, the top row shows the results for Llama-3.2-1B-It model for all three datasets. We specifically focus on the significantly high number of False Positives (25.7% in CoTemp , 32.7% in MARCO, and 71.54% in bAbI) which represent the cases where the model outputs the correct final solutions but incorrect intermediate traces. Again, the bottom row showing the results for Qwen3-1.7B model reveals similar results for the False Positive case. For both the models, we also observe a small number of True Positives which represent the cases where both the final solution and the intermediate trace are correct. For example in the cases of MARCO QA (Sub-figure 2b) and bAbI QA (Sub-figure 2c), we observe 2.5% and 22.61% True Positives against a relatively higher number of False Positives. This again highlights that a large portion of the test samples where the SFT-ed models get the correct final answer consist of incorrect intermediate traces.

Figure 3 reveals another set of striking observations from the top row showing Llama-3.2-1B-It model's results and the bottom row showing Qwen3-1.7B model's results.[2] Each of the model's confusion matrices show an alarmingly high number of False Positives across all three datasets. This represents that finetuning these models on correct solution but incorrect intermediate traces still allowed them to score high on final solution accuracy and expectedly low on intermediate trace accuracy.

As shown in the results for Table 2 and Table 3 for final solution performance, we see that SFT w/ incorrect traces outperforms SFT-Vanilla (no trace setting) and SFT w/ correct trace settings for CoTemp QA; SFT w/ correct traces does the best for Microsoft MARCO QA, while SFT-Vanilla outperformed the others in Facebook bAbI QA datasets - thereby resulting in all three possible cases strongly rejecting the hypothesis that semantic correctness of the traces does not guarantee improved final solution performance.

To further strengthen our argument, we have also conducted a $\chi^2$ statistical test with the null hypothesis: Trace accuracy and answer accuracy are independent. The test was done for both Qwen3-1.7B and Llama-3.2-1B-Instruct for the SFT w/ Incorrect trace setting. With a degree of freedom of 1, alpha=0.05, the critical value is 3.841 and we obtained $\chi^2$ to be 0.34 and 2.93 (both $< 3.841$) for the two models respectively. Hence, the null hypothesis is correct.

### 5.3 Error Analysis b/w Final Solution & Intermediate Traces

When we SFT the two language models with **correct intermediate traces and correct final solutions**, Figure 2 shows that in the cases where the final solution was incorrect (True Negatives and False Negatives), we obtained 35.15% in CoTemp QA, 27.3% in MARCO QA, and, 23.8% test samples in bAbI QA where

---

[2]In Figure 2 and Figure 3, Intermediate Trace correctness implies the cases where the models output comprises of both, the correct Classification Step output and the correct IR Step output.

incorrect final solutions were preceded by correct intermediate traces for Llama-3.2-1B-It model. Similarly, we obtained 44.32% in CoTemp QA, 45.86% in MARCO QA, and, 30.77% test samples in bAbI QA where incorrect final solutions were preceded by correct intermediate traces for Qwen3-1.7B model.

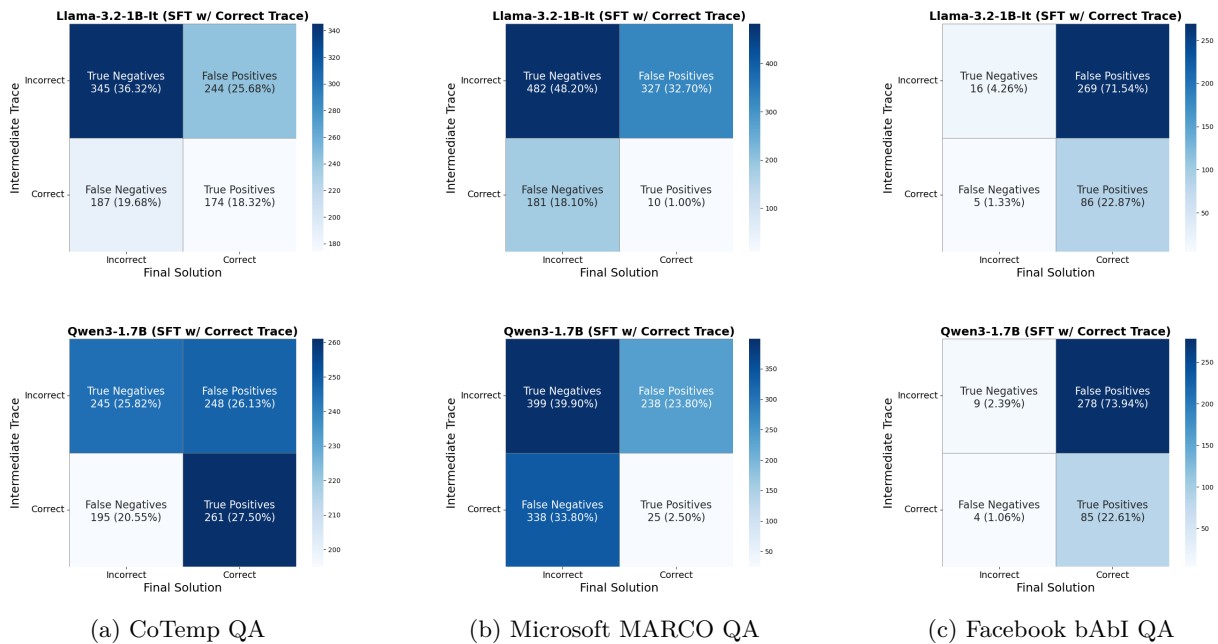

Figure 2: Confusion Matrices for **SFT w/ Correct Traces** on Llama-3.2-1B-It model (*top*) and Qwen3-1.7B (*bottom*) model, showing Final Solution Accuracy (X-axis) vs Trace Accuracy (Y-axis) for the CoTemp QA, Microsoft MARCO QA, and, the Facebook bAbI QA datasets.

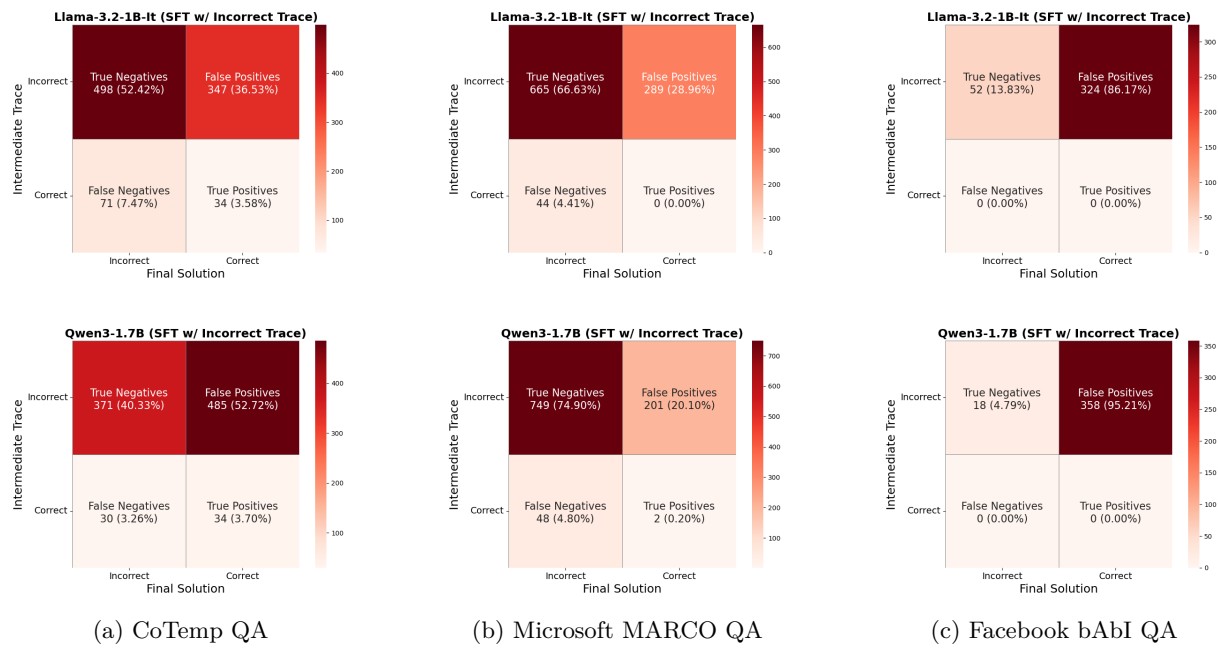

Figure 3: Confusion Matrices for **SFT w/ Incorrect Traces** on Llama-3.2-1B-It model (*top*) and Qwen3-1.7B (*bottom*) model, showing Final Solution Accuracy (X-axis) vs Trace Accuracy (Y-axis) for the CoTemp QA, Microsoft MARCO QA, and, the Facebook bAbI QA datasets.

When we look at the cases where the intermediate traces were correct, we obtained 51.8% in CoTemp QA, 94.76% in MARCO QA, and, 5.5% test samples in bAbI QA where correct intermediate traces were followed by incorrect final solutions for Llama-3.2-1B-It model. Similarly, we obtained 42.76% in CoTemp QA, 93.11% in MARCO QA, and, 4.49% test samples in bAbI QA where correct intermediate traces were followed by incorrect final solutions for Qwen3-1.7B model. In summary, our results highlight that LLMs finetuned with correct intermediate traces also led to a large number of incorrect final answers across all three QA datasets.

### 5.4 (Lack of) Correlation b/w Final Solution Accuracy & Trace Interpretability

#### 5.4.1 SFT Evaluations

We highlight the key SFT results in Figure 4. A common observation seen across three out of the four models (except in Qwen3-8B) is that SFT with R1 traces leads to the highest final solution accuracy over SFT with any other trace type with the largest performance boost seen for Llama-3.2-1B-Instruct model. Furthermore, among all the four models, we note that SFT with the algorithmically-generated semantically correct traces perform the worst also in comparison to SFT with summaries and explanations of R1 traces. Keeping these results in consideration, we conduct a user study to test if the R1 traces that led to the best performing SFT models are interpretable to humans.

#### 5.4.2 Interpretability measured through Human-Subject Studies

From Table 4, we observe that participants rated algorithmically generated correct reasoning traces as the most interpretable across all dimensions—predictability, comprehensibility, interpretability, and faithfulness—consistently scoring higher medians than all other trace types. In contrast, R1 traces received the lowest end-user interpretability ratings across every dimension. Summarized R1 traces and R1-trace explanations received intermediate ratings, indicating that compact or post-hoc representations improve human comprehension compared to raw R1 traces. In terms of cognitive workload, R1 traces imposed higher mental demand, effort, and frustration compared to other kinds of traces. Correct traces were associated with relatively lower cognitive workload, indicating that users found them easiest to follow and comprehend.

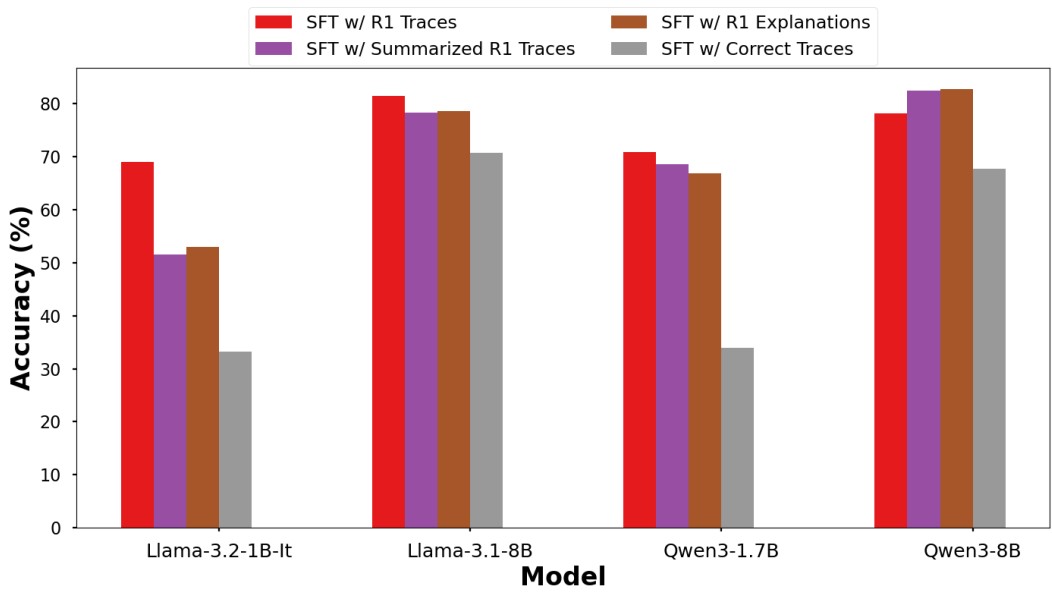

Figure 4: Final solution performance on CoTemp QA test dataset after SFT with different trace types on Llama and Qwen models.

| Dimension | Question | R1 Traces | Summarized R1 Traces | R1 Explanations | Correct Traces |
|---|---|---|---|---|---|
| Predictability | I could anticipate the next steps or conclusions based on earlier parts of the reasoning. (↑) | **3.48** | 4.45 | 4.29 | 4.82 |
| Comprehensibility | I understood the reasoning followed by the model. (↑) | **3.46** | 4.55 | 4.27 | 4.56 |
| | I could follow each step in the reasoning without confusion. (↑) | **3.46** | 4.54 | 4.28 | 4.84 |
| Interpretability | The reasoning helped me understand why the model acted or concluded the way it did. (↑) | **3.31** | 4.53 | 4.29 | 4.86 |
| Faithfulness | There were no major gaps or missing reasoning steps in the reasoning. (↑) | **3.33** | 4.54 | 4.26 | 4.72 |
| | The reasoning is consistent with the facts or evidence provided in the context. (↑) | **3.34** | 4.24 | 4.29 | 4.84 |
| Mental Demand | How mentally demanding was the task? (↓) | **4.65** | 2.87 | 2.92 | 2.31 |
| Effort | How hard did you have to work to accomplish your level of performance? (↓) | **4.54** | 2.39 | 2.17 | 2.86 |
| Frustration | How frustrated, stressed, and annoyed were you? (↓) | **4.58** | 2.04 | 2.42 | 2.42 |

Table 4: Median participant ratings of reasoning traces across dimensions of end-user interpretability and cognitive workload. Arrows indicate the desired direction of scores: ↑ higher ratings are better for interpretability measures, ↓ lower ratings are better for cognitive workload measures.

### 5.4.3 Statistical Analysis

We conducted pairwise Mann–Whitney U tests (McKnight & Najab, 2010) at a significance level of $\alpha = 0.05$ with Bonferroni correction applied for multiple comparisons. These tests were performed across all trace type pairings (R1 vs Algorithmically-generated Correct Traces, R1 vs Summarized R1 Traces, and R1 vs Post-hoc Explanations), as reported in Table 13. For the purposes of hypothesis testing, however, we focus specifically on the R1 vs algorithmically-generated Correct Traces pairing.

1. **NH-1 (Interpretability):** There is no difference in interpretability ratings between R1 traces and algorithmically-generated correct reasoning traces.

2. **NH-2 (Cognitive Workload):** There is no difference in cognitive workload ratings between R1 traces and algorithmically-generated correct reasoning traces.

There was a significant difference in end-user interpretability measured between these two trace types, across all measured dimensions (predictability: $U = 176.5$, $p = .00022 < 0.05$; comprehensibility: $U = 175$, $p = .00019 < 0.05$; interpretability: $U = 161$, $p = .00014 < 0.05$; faithfulness: $U = 178.5$, $p = .00015 < 0.05$). Further analysis also shows that there was a significant difference between cognitive workload of users between the two trace types, across all measured dimensions (mental demand: $U = 194$, $p = .00036 < 0.05$; effort: $U = 176$, $p = .00013 < 0.05$; frustration: $U = 176.5$, $p = .01287 < 0.05$). Thus, we reject both NH-1 and NH-2. Across all pairwise comparisons (Table 13), we find that R1 traces consistently differ from the other trace types in terms of end-user interpretability, with significant differences across predictability, comprehensibility, interpretability, and faithfulness. In terms of cognitive workload, R1 traces also led to significantly higher mental demand, effort, and frustration compared to correct reasoning traces. Overall, the results highlight that R1 traces degrade end-user interpretability and increase cognitive workload relative to all other reasoning traces.

### 5.5 Discussion

The key takeaways from our results can be summarized as follows:

1) *SFT w/ incorrect traces at times outperformed SFT w/ correct traces in final solution accuracy (§5.1).*

2) *Trace correctness did not guarantee final solution correctness. Solution correctness also did not imply a correct intermediate trace (§5.2 & §5.3).*

3) *Fine-tuning LLMs with the traces found to be the least interpretable by end-users led to the highest final solution accuracy, and vice-versa (§5.4).*

An important observation here is that while intermediate traces may look like how humans may approach a given problem (for example, QA in this work), they may not be the same way language models lead to generating the correct final answers. Our intervention experiments on fine-tuning the models with SFT data that consists of correct final solutions but incorrect intermediate traces provides support to this argument. On the other hand, our experiments also showed that correct answers can also be generated with incorrect traces, *which can engender a false sense of trust.* By constructing verifiable intermediate traces, we were able to demonstrate that the correctness of training traces does not even matter in the first place. This can be particularly consequential in end-user interactive systems where users are exposed to both the intermediate traces and the final solutions generated by the models.

Our findings from the end-user interpretability vs performance trade-off experiments reveal a significant disconnect between the utility of reasoning traces for improving LLM performance and their cognitive interpretability for humans. SFT with R1 traces led to higher accuracy, yet these traces were rated the lowest across all dimensions of end-user interpretability—predictability, comprehensibility, and faithfulness—and were associated with the highest mental demand, effort, and frustration. Summaries and post-hoc explanations of R1 traces further validate this point: although they yielded lower accuracy than R1 traces, they are easier for users to predict, understand, and perceive as faithful (but also lack systematic evaluation). By contrast, algorithmically generated correct traces were judged as most interpretable and least mentally demanding, but yielded the weakest improvements in model accuracy. These results clearly highlight that verbose traces like R1 provide rich training signals for models, but are poorly aligned with the interpretability expectations of the end user. Furthermore, the reasoning traces which benefit the LLMs the most need not have semantic structure, underscoring a fundamental disconnect between what serves as a good training signal and what supports human understanding.

**Scope:** OpenBook QA domain is not only representative of a critical application of today's interactive dialogue systems, but more importantly offers a controlled experiment environment for demonstrating the disconnect between trace fidelity and final accuracy. In other reasoning areas, as shown by commonly used test datasets like AIME/MATH500 for math and LiveCodeBench for code, there are many problems where creating a step-by-step reasoning process that works for every math or code problem is not possible. This creates a problem for evaluating the intermediate steps in these situations. The main point of this work is to demonstrate a gap between the accuracy of the steps and the final result for OpenBook QA problems. We also note that there are other contemporaneous works that seem to show a similar disconnect between intermediate tokens and final solution accuracy in the context of planning-type scenarios (Stechly et al., 2025).

## 6  Conclusion

In this work, we investigate the correlation between an LLM's task performance, i.e., final solution accuracy and two properties of the intermediate traces generated after SFT, namely - semantic correctness and end-user interpretability. By using a Knowledge Distillation method which utilizes problem decomposition making these intermediate traces objectively verifiable, we evaluated the correctness of both the final solution and the intermediate traces at the time of inference, specifically for Open Book QA reasoning problems. Our SFT experiments with correct and incorrect intermediate traces on Llama and Qwen models across QA datasets reveal that there is no significant correlation between final solution accuracy and intermediate trace accuracy. In our SFT experiments with R1 traces, their summaries, and post-hoc explanations generated

using GPT-4o-mini, we saw that using R1 traces achieved the highest final solution accuracy. However, from our human-subject study we found that R1 traces were rated lowest in all human interpretability metrics, revealing that traces most beneficial for the model often impose the greatest cognitive burden on users. This decoupling between model performance and trace interpretability leads to two main takeaways: (1) verbose trace formats like CoT seems to be helpful for improving model performance rather than end-user explanations, and (2) generating human-interpretable rationales must be addressed independently via dedicated explanation modules or a separate training objective. Our results highlight the need for a more nuanced approach in designing trace-based training/fine-tuning methods and explanation systems for future LLMs, balancing both LLM performance and user understanding.

## Broader Impact

With the surge in number of end-user interactions with dialogue systems such as ChatGPT, Perplexity, Microsoft Copilot or Google Gemini, there is also a growing need to deploy SLMs which provide computationally efficient alternatives but lack the performance of Large Language Models. Task-specific SFT, and more recently with Input-Trace-Output tuples, has shown improved final solution performance for these SLMs. However, since these systems directly interface with end users who expect accuracy and transparency in the models' outputs, our work points at the lack of verifiable and interpretable intermediate traces being used for finetuning language models. We also highlight the consequential impact of a model outputting intermediate 'reasoning' traces which hold little to no correlation with correct final solutions. We believe that our findings motivate future works to independently design solutions for boosting LLM performance and for generating post-hoc explanations for end-users.

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

# Contents

# A Additional Experiment Details

## A.1 Dataset Distributions

Table 5: Train and Test data distribution for CoTemp QA dataset used in our SFT experiments.

| Category | Train/Test Samples |
|---|---|
| equal | 349 / 87 |
| overlap | 522 / 131 |
| during | 2477 / 619 |
| mix | 450 / 113 |

Table 6: Train and Test data distribution for Microsoft MARCO QA dataset used in our SFT experiments.

| Category | Train/Test Samples |
|---|---|
| description | 1000 / 200 |
| entity | 1000 / 200 |
| numeric | 1000 / 200 |
| location | 1000 / 200 |
| person | 1000 / 200 |

Table 7: Train and Test data distribution for Facebook bAbI QA dataset used in our SFT experiments.

| Category | Train/Test Samples |
|---|---|
| single-supporting-fact | 200 / 20 |
| two-supporting-facts | 200 / 20 |
| two-arg-relations | 1000 / 100 |
| counting | 200 / 20 |
| list-sets | 200 / 20 |
| conjunction | 200 / 20 |
| time-reasoning | 200 / 20 |
| basic-deduction | 250 / 25 |
| basic-induction | 1000 / 100 |
| positional-reasoning | 125 / 12 |
| size-reasoning | 198 / 19 |

## A.2    Prompts

---

**R1 Trace Summarization Prompt**

Summarize the following trace in a very concise and clear manner, highlighting key events and outcomes in less than 100 words:

...

**{R1 trace}**

...

**Summary:**

---

**R1 Trace Explanation Prompt**

**{Problem}**

...

**{R1 trace}**

...

**{R1 answer}**

You have answered the question correctly. Please provide a detailed explanation of the reasoning behind your answer. The explanation should be clear, concise, and easy to understand.

...

**Explanation:**

---

## A.3  Trace Length Analysis

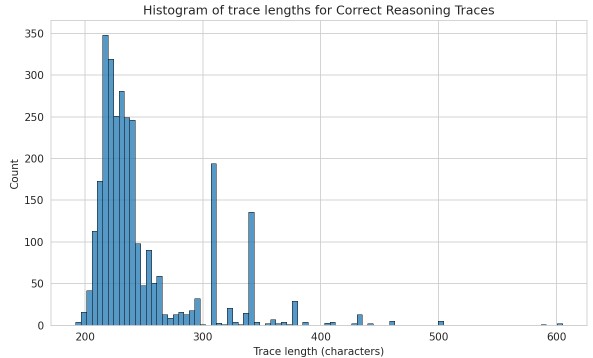

(a) Distribution of trace lengths of algorithmically generated correct traces.

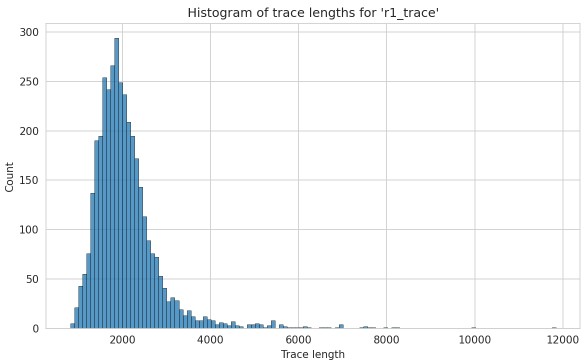

(b) Distribution of trace lengths of R1 traces

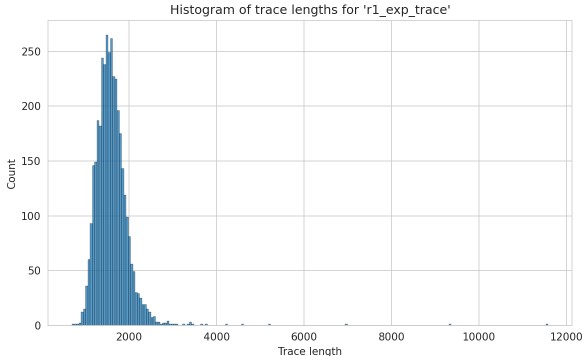

(c) Distribution of trace lengths of post-hoc explanations of R1 traces

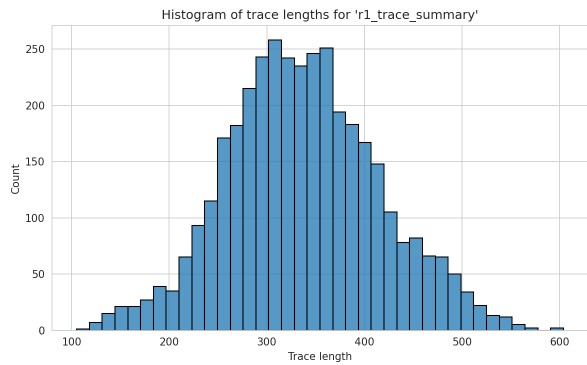

(d) Distribution of trace lengths of summaries of R1 traces

Figure 5: Histogram comparison of trace lengths for the four reasoning trace types used in our study: (a) algorithmically generated correct traces (b) R1 chain-of-thought traces (c) post-hoc explanations of R1 traces (d) summarized R1 traces

# B   User Study

To evaluate the interpretability of reasoning traces generated by reasoning models, we conducted a set of structured user studies. Each participant was given a compensation of $12/$hr$. The IRB protocol details will be released on acceptance. Each study followed the same sequence of steps, designed to ensure consistency across participants for each study. Below we outline the main components of the study design.

## B.1   Human Participant Demographics

We conducted four user studies with participants recruited through Prolific (all located in the United States). In general, the participant populations in all four studies were demographically similar, with no major differences in the age or education distribution, suggesting that the results in the studies are comparable and not driven by differences in the composition of the participants.

**Education:**   Participants spanned a range of educational backgrounds. Across all studies, the majority held an *Undergraduate Degree* (roughly 45–55% in each study), followed by *Master's Degrees* (20–30%), and a smaller proportion with *PhDs or equivalent doctoral-level degrees* (10–15%). A minority of participants reported *High School*, *Associate's Degree*, or *Some College* as their highest level of education (<10% each). These proportions were consistent across the four studies.

**Age:**   The participants were distributed over a wide age range, with the largest groups being *35–50 years old* (approximately 35–40%) and *51+ years old* (30–35%). Younger age groups were represented to a lesser extent: *26–34 years old* (20–25%) and *18–25 years old* (5–10%). Again, these proportions were stable across studies.

## B.2   Consent and Statement

Each participant began the study by reviewing and agreeing to a consent statement (Figure 6. The statement explained the goals of the study, what participants would be asked to do, and how their data would be handled.

## User Study

**What to Expect**
This study has been reviewed and approved by the Institutional Review Board (IRB) with code:55564726. During the study, you will be presented with five passage-based questions. We expect the study to take approximately 15 minutes to complete.

**Confidentiality**
All data collected during this study will be kept strictly confidential. Your personal information will remain anonymous, and will only be accessible by authorized research investigators. The information will only be used for research purposes and will not be shared with any external entities.

**Prolific ID**
At the end of the study (when you click "Submit") you will be provided Prolific Remuneration Code. Please
retain that code for your reference and enter it in Prolific Interface to get the remuneration.

Figure 6: Consent statement shown to participants before starting the study.

### B.3 Instructions

Participants were provided with detailed instructions describing the study structure (Figure 7). Each of the five parts of the study followed the same format:

1. **Facts:** A short list of factual statements about a person.

2. **Question:** A query based on the passage.

3. **Model's Answer:** The response generated by the AI model.

4. **Reasoning:** A step-by-step explanation of how the model arrived at its answer.

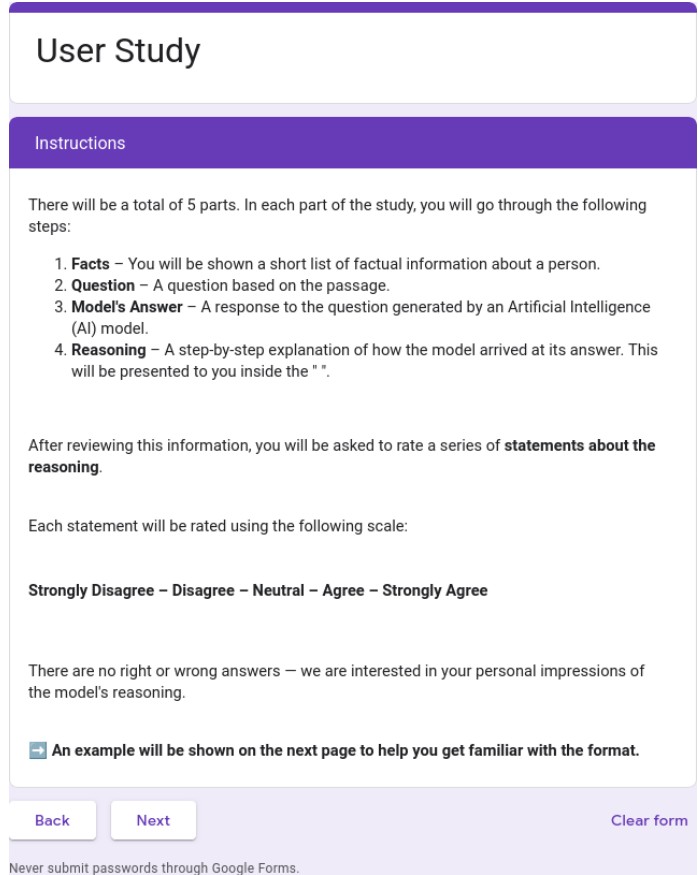

Figure 7: Instructions shown to participants before starting the study.

After reviewing this information, participants rated statements about the reasoning on a 5-point Likert scale (Strongly Disagree–Strongly Agree).

### B.4 Q/A Task

Before beginning the main task, participants reviewed an example question and answer with reasoning (Figure 8). Participants then completed five Q/A tasks of the same form as the example. Each task included a passage, model answer, reasoning trace, and associated questionnaire (Figure 9).

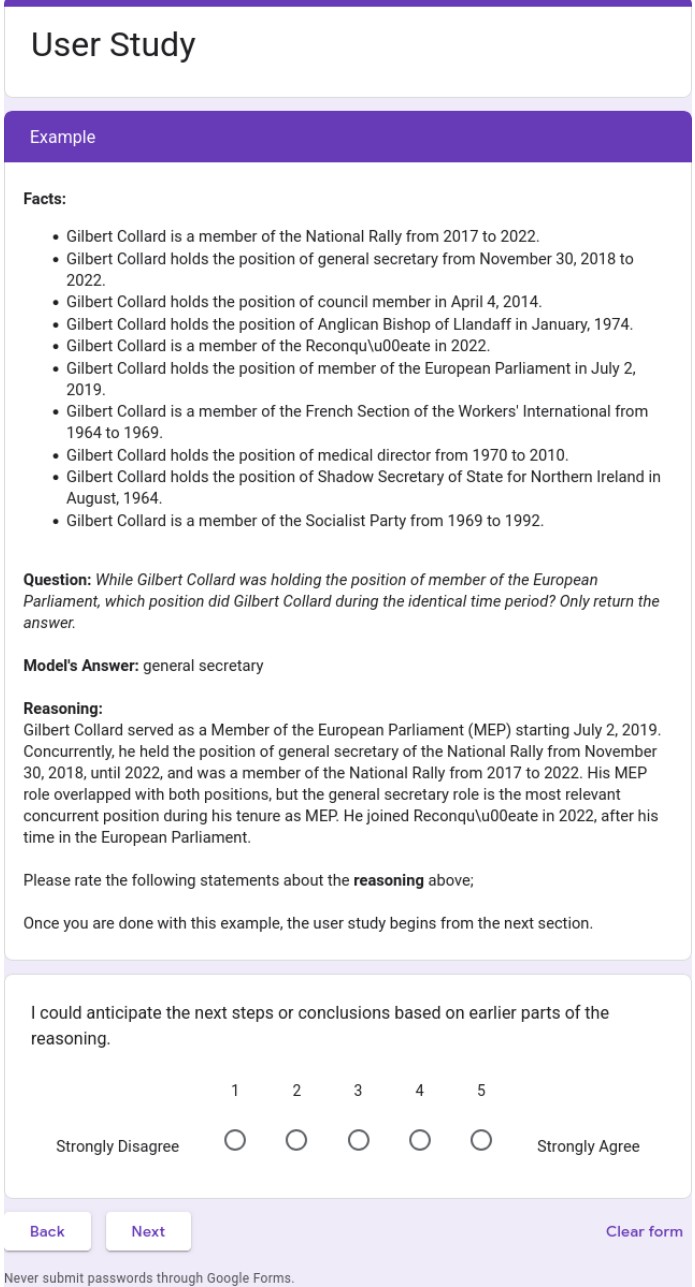

Figure 8: Example shown to participants.

**User Study**

**Question 1**

**Facts:**

- Stine Bosse worked for Paramount Pictures from January 14, 1992 to November 24, 1999.
- Stine Bosse worked for IBM Almaden Research Center in October, 2008.
- Stine Bosse attended University of Copenhagen in 1987.
- Stine Bosse worked for Thomas Edison State University from June, 2007 to January, 2011.
- Stine Bosse works for Tryg from September 30, 2002 to February 1, 2011.
- Stine Bosse works for TDC from September 27, 2004 to February 28, 2006.
- Stine Bosse works for Alka from September 30, 2002 to February 1, 2011.
- Stine Bosse worked for National University of Science and Technology in January, 1999.
- Stine Bosse worked for Vassar College from March 26, 1998 to March 28, 1999.
- Stine Bosse works for Nykredit from January 16, 1989 to January 28, 1992.

**Question:** *While Stine Bosse was working for TDC, which employer did Stine Bosse work for within the same time interval? Only return the answer.*

**Model's Answer:** Tryg and Alka

**Reasoning:**
Stine Bosse worked for TDC from September 27, 2004, to February 28, 2006. During this period, she also worked for Tryg and Alka, both of which had overlapping employment dates from September 30, 2002, to February 1, 2011. Although the question asks for a singular employer, both Tryg and Alka qualify as valid answers.

Please rate the following statements about the **reasoning** above:-

Figure 9: Task shown to participants.

## B.5 Examples of Traces shown to Users

Table. 8, 9, 10, 11, 12 compares four representations of reasoning traces used in the study: the full R1 traces, summarized traces, post-hoc generated explanations, and the correct traces.

Table 8: Comparison of R1 traces, summarized traces, post-hoc explanations, and correct traces for interpretability analysis: Instance 1

| Type of Trace | Trace generated by model after SFT |
| --- | --- |
| DeepSeek R1 | "Hmm, the user is asking about Gilbert Collard's positions during the time he was a member of the European Parliament. Let me carefully analyze the context provided. Looking at the timeline, Gilbert Collard became a member of the European Parliament on July 2, 2019. I need to find which other positions he held simultaneously during this period. From the context: He was general secretary of National Rally from November 30, 2018 to 2022. He was a member of National Rally from 2017 to 2022. He joined Reconquête in 2022. Since his European Parliament term started in July 2019 and overlapped with both his National Rally membership (2017-2022) and his general secretary position (2018-2022), both positions were held concurrently with his MEP role. However, the question specifically asks for positions held during the identical time period as his MEP tenure. Both 'member of National Rally' and 'general secretary' qualify, but I need to see which one is more precise. The general secretary role started before his MEP term (Nov 2018 vs July 2019) and ended at the same time (2022). The party membership also covers the entire period. But since the question mentions 'position,' both are valid, though general secretary is a specific appointed role. I notice the user's question says 'which position' (singular), so I should provide the most relevant concurrent position. Given that general secretary is an active leadership role within the party he belonged to during his MEP term, it seems the most directly relevant answer. The other positions either predate his MEP term (like his medical director role ending in 2010) or are from different periods (like his Socialist Party membership ending in 1992). The Anglican Bishop and Shadow Secretary positions seem like possible data errors as they don't align with his known political career. Therefore, while holding the MEP position from July 2019 onward, Gilbert Collard was simultaneously the general secretary of National Rally until 2022." |
| LLM-generated R1 Summary | "Gilbert Collard served as a Member of the European Parliament (MEP) starting July 2, 2019. Concurrently, he held the position of general secretary of the National Rally from November 30, 2018, until 2022, and was a member of the National Rally from 2017 to 2022. His MEP role overlapped with both positions, but the general secretary role is the most relevant concurrent position during his tenure as MEP. He joined Reconquête in 2022, after his time in the European Parliament." |

Table 8 – continued from previous page

| Type of Trace | Trace generated by model after SFT |
| --- | --- |
| LLM-generated R1 post-hoc Explanation | "Alright, let me break down how I arrived at the answer step by step. The user wants to know which position Gilbert Collard held simultaneously while serving as a Member of the European Parliament (MEP). First, I identify the key timeframe: Collard became an MEP on July 2, 2019. This is my anchor point. Now I need to examine all his other positions to see which overlapped exactly with this period. Looking through the data: His role as general secretary ran from November 30, 2018 to 2022. Since July 2019 falls between these dates, this overlaps. His National Rally membership (2017-2022) also covers the MEP period, but the question specifically asks for 'position' - implying a formal role, not just membership. Other positions either ended before 2019 (like medical director ending in 2010) or started after he left the MEP role (Reconquête membership in 2022). Notably, some entries seem implausible based on known facts (like Anglican Bishop in 1974 or Shadow Secretary in 1964), so I disregard those as likely errors. Therefore, the only verifiable formal position held concurrently with his MEP term is general secretary of National Rally. The timeline alignment is clear: he held both roles continuously from July 2019 until 2022." |
| Verifiably Correct Trace | "The temporal relation between the event in question and the event in context is: during. I need to use the following facts to answer the question: {'Gilbert Collard holds the position of general secretary from November 30, 2018 to 2022.'}" |
| Verifiably Incorrect Trace | "The temporal relation between the event in question and the event in context is: equal.I need to use the following facts to answer the question: Gilbert Collard is a member of the Socialist Party from 1969 to 1992." |

Table 9: Comparison of R1 traces, summarized traces, post-hoc explanations, and correct traces for interpretability analysis: Instance 2

| Type of Trace | Trace generated by model after SFT |
| --- | --- |
| DeepSeek R1 | "Okay, I'm trying to figure out this question about Stine Bosse's work history. The question is asking which employer she worked for at the same time she was working for TDC. Let me look at the timeline carefully.First, I see that Stine worked for TDC from September 27, 2004 to February 28, 2006. So that's the time period we're focusing on. Now I need to check which other employers she worked for during this exact timeframe.Looking at the list:Paramount Pictures was from 1992 to 1999, which is before TDC.IBM Almaden was in October 2008, after TDC.University of Copenhagen was in 1987, way before.Thomas Edison State University was from 2007-2011, after TDC.Tryg was from 2002-2011, which overlaps with TDC's timeframe.Alka was also from 2002-2011, overlapping too.National University of Science and Technology was in January 1999, before.Vassar College was from 1998-1999, also before.Nykredit was from 1989-1992, way before.So both Tryg and Alka have employment periods that completely cover the TDC timeframe. Since the question asks which employer she worked for "within the same time interval" as TDC, both Tryg and Alka qualify.But I notice something interesting - both Tryg and Alka have identical employment dates: September 30, 2002 to February 1, 2011. This exactly encompasses the TDC period.The question says "which employer" (singular) but technically there are two overlapping employers. However, since both are valid answers and the context doesn't specify choosing only one, I should list both. But the instruction says "only return the answer" so I'll provide both names since they both meet the criteria." |
| LLM-generated R1 Summary | "Stine Bosse worked for TDC from September 27, 2004, to February 28, 2006. During this period, she also worked for Tryg and Alka, both of which had overlapping employment dates from September 30, 2002, to February 1, 2011. Although the question asks for a singular employer, both Tryg and Alka qualify as valid answers." |
| LLM-generated R1 post-hoc Explanation | "Alright, let me break this down. The user wants a clear explanation of how I determined which employers Stine Bosse worked for simultaneously with TDC. First, I recall that the key task was cross-referencing employment periods. The critical dates are TDC's timeframe: September 27, 2004, to February 28, 2006. I scanned all other employers: Paramount Pictures (1992-1999) ended before TDC started. IBM (2008) and Thomas Edison (2007-2011) started after TDC ended. University of Copenhagen (1987) and Nykredit (1989-1992) were too early. National University (Jan 1999) and Vassar College (1998-1999) didn't overlap. But Tryg and Alka both spanned 2002-2011—fully encompassing TDC's duration. The user seems satisfied with the initial answer but wants deeper reasoning. They might be testing my ability to logically parse timelines or preparing for similar data analysis tasks. What's interesting is how Tryg/Alka share identical dates. This suggests either parallel employment or data entry quirks, but the question only cares about temporal overlap—which both satisfy. I'll emphasize the date comparisons clearly, avoiding jargon. Since the user appreciated conciseness before, I'll keep the explanation tight but thorough." |
| Verifiably Correct Trace | "The temporal relation between the event in question and the event in context is: during. I need to use the following facts to answer the question: ['Stine Bosse works for Tryg from September 30, 2002 to February 1, 2011.', 'Stine Bosse works for Alka from September 30, 2002 to February 1, 2011.']." |
| Verifiably Incorrect Trace | "The temporal relation between the event in question and the event in context is: mix.I need to use the following facts to answer the question: Stine Bosse worked for Thomas Edison State University from June, 2007 to January, 2011." |

Table 9 – continued from previous page

| Type of Trace | Trace generated by model after SFT |
| --- | --- |
| | |

Table 10: Comparison of R1 traces, summarized traces, post-hoc explanations, and correct traces for interpretability analysis: Instance 3

| Type of Trace | Trace generated by model after SFT |
| --- | --- |
| DeepSeek R1 | "Alright, so I need to figure out which position Thomas Wenski held at the same time he was the auxiliary bishop. Let me start by looking at the context provided.First, I see that Thomas Wenski became the auxiliary bishop on June 24, 1997. That's a specific date, so I should look for other positions he held around that same time. Scanning through the list, I notice several other positions with dates:Titular bishop on June 24, 1997.Catholic bishop on September 3, 1997.Minister for Children (Denmark) on June 2, 1998.Head of the Security Service of Ukraine on April 13, 1999.Director of the Federal Security Service in May 1999.President of Vietnam from 1999 to 2001.Deputy President of the Supreme Court of the UK in November 2002.Parliamentary Under-Secretary of State for Employment from November to December 2004.Diocesan bishop on November 13, 2004.Wait a minute, I see that both the auxiliary bishop and titular bishop positions started on the exact same date: June 24, 1997. That means that while he was holding the position of auxiliary bishop, he was also the titular bishop at the same time. The other positions have different dates, either before or after June 24, 1997, so they wouldn't overlap. For example, the Catholic bishop role started a few months later in September, and the Minister for Children role started almost a year later in June 1998. All the other positions are even later, so they don't coincide with his time as auxiliary bishop.Therefore, the position Thomas Wenski held during the same time period as auxiliary bishop is the titular bishop." |
| LLM-generated R1 Summary | "Thomas Wenski became auxiliary bishop on June 24, 1997, simultaneously holding the position of titular bishop, which also started on the same date. Other roles he held, such as Catholic bishop and Minister for Children, began later, indicating no overlap with his auxiliary bishop tenure. Thus, the only concurrent position was titular bishop." |

Table 10 – continued from previous page

| Type of Trace | Trace generated by model after SFT |
| --- | --- |
| LLM-generated R1 post-hoc Explanation | "Okay, the user wants me to explain my reasoning for the previous answer about Thomas Wenski holding positions simultaneously. I correctly identified that he was both auxiliary bishop and titular bishop on the same start date.Hmm, looking at the context again, every position entry has clear dates. The key was noticing that both "auxiliary bishop" and "titular bishop" shared the exact start date of June 24, 1997. That's the critical overlap.The user seems satisfied with the answer but wants a clear breakdown. They're probably verifying if I understood the temporal reasoning or just guessed. Maybe they're testing how I handle simultaneous appointments.I should emphasize the date matching in the explanation. Other positions either started later (like Catholic bishop in September) or much later (1998-2004), so they couldn't overlap with the June 1997 auxiliary bishop role.The explanation should stay concise but highlight why only titular bishop fits. No need to mention all other positions - just contrast with one nearby example (like September's Catholic bishop role) to show the time gap.User's phrasing suggests they value precision. They said "during the same time period", so I'll reinforce that it's about exact date coincidence, not just vague overlapping periods....Wait, is there any chance these bishop titles could be sequential? No - the identical start dates confirm concurrency. Ecclesiastical roles often work that way.Final explanation structure:State the key dateShow titular bishop matchContrast with closest non-matchConclude exclusively. Keep it under 3 sentences." |
| Verifiably Correct Trace | "The temporal relation between the event in question and the event in context is: equal. I need to use the following facts to answer the question: ['Thomas Wenski holds the position of titular bishop in June 24, 1997.']" |
| Verifiably Incorrect Trace | "The temporal relation between the event in question and the event in context is: overlap. I need to use the following facts to answer the question: Thomas Wenski holds the position of auxiliary bishop in June 24, 1997." |

Table 11: Comparison of R1 traces, summarized traces, post-hoc explanations, and correct traces for interpretability analysis: Instance 4

| Type of Trace | Trace generated by model after SFT |
| --- | --- |
| DeepSeek R1 | "Alright, so I need to figure out which position Kalervo Kummola held while he was the vice president. Let me break this down step by step to make sure I get it right.First, I'll look at the timeline for when he was vice president. The context says that Kalervo Kummola was the vice president from 2003 to 2021. That's a pretty long period—almost 18 years. So, any position he held during this time frame would be the answer.Now, I'll go through each of the positions listed in the context and check their dates to see which ones overlap with his vice presidency period.President of Romania in 2002: This is before 2003, so it doesn't overlap with his vice presidency.Foreign minister in November 1998: Also before 2003, so no overlap.United States Deputy Secretary of Labor in 1999: Before 2003, so no.Minister for Nordic Co-operation in July 1998: Again, before 2003.Anglican Bishop of Salisbury in 1999: Still before 2003.Confederate States Secretary of State in September 20, 2001: This is in 2001, which is before 2003.Lord Lieutenant of Ireland in August 1999: Before 2003.Board member from 1998 to 2021: This position starts in 1998 and ends in 2021. Since his vice presidency is from 2003 to 2021, there's an overlap from 2003 to 2021.Chairperson from 1997 to January 7, 2016: This ends in January 2016, which is within the vice presidency period (2003-2021). So, he was chairperson until 2016 while being vice president.Vice president from 2003 to 2021: This is the position we're comparing against.So, looking at the positions that overlap with his vice presidency:Board member: From 1998 to 2021. Since the vice presidency starts in 2003, the overlap is from 2003 to 2021.Chairperson: From 1997 to January 7, 2016. This overlaps with his vice presidency from 2003 until January 2016.Both positions overlap with his vice presidency period. However, the question asks which position he held within the same time interval as being vice president. Both board member and chairperson fit this criterion. But perhaps I need to see if there's a more specific answer or if both are acceptable.Wait, the context lists multiple positions, but the question is asking for which position he held while he was vice president. Since both board member and chairperson were held concurrently with his vice presidency, I need to list both. But the instruction says "Only return the answer," so I should present both positions clearly.Alternatively, maybe the question expects the positions that were held exclusively during the vice presidency, but that might not be the case since both roles span beyond the start and end of his vice presidency but still overlap significantly.To be precise, both board member and chairperson were positions he held during his vice presidency. So, the answer should include both." |
| LLM-generated R1 Summary | "Kalervo Kummola served as vice president from 2003 to 2021. During this period, he also held the positions of board member (1998-2021) and chairperson (1997-January 2016). Both roles overlapped with his vice presidency, making them relevant to the inquiry. Thus, the answer includes both positions." |

**Table 11 – continued from previous page**

| Type of Trace | Trace generated by model after SFT |
|---|---|
| LLM-generated R1 post-hoc Explanation | "Okay, the user wants me to explain my reasoning behind identifying which positions Kalervo Kummola held while serving as vice president. Let me recall the context clearly: the vice presidency spanned from 2003 to 2021, and I need to find overlapping roles.First, I'll mentally scan all listed positions. Most roles like President of Romania (2002) or foreign minister (1998) clearly fall outside the 2003-2021 window. The Confederate States Secretary role in 2001 is too early too.Then two positions catch my eye: board member (1998-2021) and chairperson (1997-Jan 2016). Both end dates extend into the vice presidency period. For board member, the entire 2003-2021 duration is covered under its 1998-2021 tenure. For chairperson, even though it started earlier, it actively overlapped from 2003 until January 2016—which is well within the vice presidency timeline.The user seems satisfied with the initial answer but wants deeper clarity. They might be testing if I understand temporal overlaps precisely, or perhaps preparing for similar logic puzzles. Since all other positions are time-bound snapshots (single years/months) that don't touch 2003-2021, only these two long-term roles qualify.I should emphasize why the other roles don't fit—like how "July 1998" or "1999" are discrete points years before 2003—to prevent ambiguity. The explanation must stay concise but airtight, as the user values precision." |
| Verifiably Correct Trace | "The temporal relation between the event in question and the event in context is: mix. I need to use the following facts to answer the question: ['Kalervo Kummola holds the position of board member from 1998 to 2021.', 'Kalervo Kummola holds the position of chairperson from 1997 to January 7, 2016.']" |
| Verifiably Incorrect Trace | "The temporal relation between the event in question and the event in context is: equal. I need to use the following facts to answer the question: Kalervo Kummola holds the position of President of Romania in 2002." |

Table 12: Comparison of R1 traces, summarized traces, post-hoc explanations, and correct traces for interpretability analysis: Instance 5

| Type of Trace | Trace generated by model after SFT |
| --- | --- |
| DeepSeek R1 | ""Alright, let's tackle this problem step by step. The question is asking which position Pierre-Joseph-Olivier Chauveau held simultaneously while he was a member of the House of Commons of Canada. So, first, I need to figure out the timeframe during which he held that position.Looking at the context provided, Pierre-Joseph-Olivier Chauveau was a member of the House of Commons of Canada from September 20, 1867, to January 22, 1874. That's a period of about six and a half years. Now, I need to see which other positions he held during this same timeframe.Let's list out all the positions he held and their respective dates:Speaker of the New Zealand House of Representatives: July 1874Speaker of the Senate: February 21, 1873, to January 8, 1874Catholic bishop: March 1867President of Singapore: May 9, 1874Prime Minister of Albania: August 1874Prime Minister of New Zealand: October 11, 1874Minister of Works: February 1874Member of the House of Commons of Canada: September 20, 1867, to January 22, 1874Secretary of Education: June 9, 1867Premier of Quebec: July 15, 1867, to February 25, 1873Now, I'll compare each position's dates to his tenure in the House of Commons. The House of Commons role ended on January 22, 1874. So any position that overlaps with this timeframe would be simultaneous.Speaker of the New Zealand House of Representatives: July 1874. Since his House of Commons term ended in January 1874, this position starts after that, so no overlap.Speaker of the Senate: February 21, 1873, to January 8, 1874. This ends just before his House of Commons term ends, so there's an overlap here. He held both positions from February 21, 1873, to January 8, 1874.Catholic bishop: March 1867. This is before his House of Commons term even started (which began in September 1867), so no overlap.President of Singapore: May 9, 1874. This is after his House of Commons term ended, so no overlap.Prime Minister of Albania: August 1874. Also after his term ended, so no overlap.Prime Minister of New Zealand: October 11, 1874. After his term ended, no overlap.Minister of Works: February 1874. This is within his House of Commons term, which ended in January 22, 1874. Wait, February 1874 is after January 22, 1874? Let me double-check the dates. If his House of Commons term ended on January 22, 1874, then February 1874 is after that, so no overlap.Secretary of Education: June 9, 1867. This is before his House of Commons term started in September 1867, so no overlap.Premier of Quebec: July 15, 1867, to February 25, 1873. This starts before his House of Commons term and ends during it. Specifically, it ends on February 25, 1873, while his House of Commons term continues until January 22, 1874. So, there's an overlap here from July 15, 1867, to February 25, 1873.So, based on this analysis, Pierre-Joseph-Olivier Chauveau simultaneously held the position of Premier of Quebec while he was a member of the House of Commons of Canada. Additionally, he also simultaneously held the position of Speaker of the Senate for a brief period from February 21, 1873, to January 8, 1874.Therefore, the positions he held simultaneously while being a member of the House of Commons of Canada are Premier of Quebec and Speaker of the Senate."" |
| LLM-generated R1 Summary | "Pierre-Joseph-Olivier Chauveau served as a member of the House of Commons of Canada from September 20, 1867, to January 22, 1874. During this period, he simultaneously held the positions of Premier of Quebec (July 15, 1867 - February 25, 1873) and Speaker of the Senate (February 21, 1873 - January 8, 1874). Thus, his overlapping roles were as Premier of Quebec and Speaker of the Senate while serving in the House of Commons." |

**Table 12 – continued from previous page**

| Type of Trace | Trace generated by model after SFT |
|---|---|
| LLM-generated R1 post-hoc Explanation | "Alright, the user wants a detailed explanation of how I arrived at the correct answer. Let me start by recalling the original question: which position did Pierre-Joseph-Olivier Chauveau hold simultaneously while being a member of the House of Commons of Canada?First, I need to establish the timeframe for his House of Commons membership. The context states it was from September 20, 1867, to January 22, 1874. That's about six and a half years. Now, I have to check all other positions listed and see which ones overlap with this period.Looking at each position:Speaker of the New Zealand House of Representatives (July 1874): This is after January 1874, so no overlap.Speaker of the Senate (Feb 21, 1873 - Jan 8, 1874): This ends just before his House term ends. Overlap from Feb 1873 to Jan 1874.Catholic bishop (March 1867): Before his House term started. No overlap.President of Singapore (May 9, 1874): After his term. No.Prime Minister of Albania (August 1874): Also after. No.Prime Minister of New Zealand (October 11, 1874): Again, after. No.Minister of Works (February 1874): His House term ended in January, so February is after. No overlap.Secretary of Education (June 9, 1867): Before September 1867. No.Premier of Quebec (July 15, 1867 - February 25, 1873): This starts before his House term and ends during it. Overlap from July 1867 to Feb 1873.So, both Premier of Quebec and Speaker of the Senate overlap. But the user's answer only mentioned Premier of Quebec. Hmm. Let me double-check the Speaker of the Senate dates. It started in Feb 1873, while he was still in the House (until Jan 1874). So technically, he held both during that period. But the user's answer didn't include it. Maybe the question is about positions held for the entire duration? Or perhaps the user considers only the primary position?Wait, the original answer provided by the assistant was "Premier of Quebec." The user confirmed it was correct. So why didn't the assistant include Speaker of the Senate? Let me see the overlap period again. His House term ended Jan 22, 1874. The Speaker role ended Jan 8, 1874. So yes, there's an overlap of about two weeks in January 1874. But maybe the assistant considered the main concurrent role as Premier, which overlapped for a much longer period (over 5 years vs. less than a year for Speaker). Or perhaps the context implies that Speaker of the Senate is a different legislative body, making it impossible to hold both? But technically, one is House of Commons (lower) and Senate (upper)—so in some systems, holding both isn't feasible.Ah, that's likely the key. In Canada, the House of Commons and Senate are separate chambers. A person can't simultaneously be a member of both. So Pierre couldn't be an MP (House of Commons) and Speaker of the Senate at the same time. The dates might overlap, but constitutionally, it's impossible. Hence, only Premier of Quebec is valid as a simultaneous role since provincial premiership and federal MP roles can coexist. That explains why the assistant correctly listed only Premier of Quebec.So the reasoning hinges on understanding the compatibility of roles. The dates alone might suggest two overlaps, but real-world constraints (like not serving in two legislative bodies) make only one valid. The user probably expected this nuance. Good to confirm!" |
| Verifiably Correct Trace | "The temporal relation between the event in question and the event in context is: mix. I need to use the following facts to answer the question: ['Pierre-Joseph-Olivier Chauveau holds the position of Speaker of the Senate from February 21, 1873 to January 8, 1874.', 'Pierre-Joseph-Olivier Chauveau holds the position of Premier of Quebec from July 15, 1867 to February 25, 1873.']" |
| Verifiably Incorrect Trace | "The temporal relation between the event in question and the event in context is: during. I need to use the following facts to answer the question: Pierre-Joseph-Olivier Chauveau holds the position of Prime Minister of Albania in August, 1874." |

| Table 12 – continued from previous page | |
|---|---|
| **Type of Trace** | **Trace generated by model after SFT** |

## B.6 Questionnaire

After each reasoning trace, participants filled out a brief questionnaire assessing dimensions of interpretability (Figure 10). At the end of the study, participants completed a feedback survey, with NASA-TLX questions to measure perceived workload (Figure 11).

Figure 10: Task-specific questionnaire shown to participants.

Figure 11: Final screen shown to participants for NASA TLX questions.

### B.7 Statistical Analysis Results

| Dimension | R1 vs Correct | | | R1 vs Summarized R1 | | | R1 vs Explanations | | |
|---|---|---|---|---|---|---|---|---|---|
| | $U$ | $p$ | Sig. | $U$ | $p$ | Sig. | $U$ | $p$ | Sig. |
| **H1: Interpretability** | | | | | | | | | |
| Predictability | 176.5 | .00022 | Yes | 177.0 | .00036 | Yes | 126.5 | .0004 | Yes |
| Comprehensibility | 175 | .00019 | Yes | 102.2 | $< .00001$ | Yes | 126 | .0006 | Yes |
| Interpretability | 161 | .00014 | Yes | 74.5 | $< .00001$ | Yes | 187 | $< .00001$ | Yes |
| Faithfulness | 178.5 | .00015 | Yes | 73.5 | $< .00001$ | Yes | 115.5 | $< .00001$ | Yes |
| **H2: Cognitive Workload** | | | | | | | | | |
| Mental Demand | 194 | .00036 | Yes | 237.5 | 0.055 | No | 264 | .21 | No |
| Effort | 176 | .00013 | Yes | 169.5 | .0016 | Yes | 104 | $< .00001$ | Yes |
| Frustration | 102.5 | .01287 | Yes | 164 | 0.0013 | Yes | 158 | .00056 | Yes |

Table 13: Pairwise Mann–Whitney U test results across different trace types. Significance is determined at $\alpha = 0.05$ with Bonferroni correction.

