# OpenReview forum: "Interpretable Traces, Unexpected Outcomes: Investigating the Disconnect in Trace-Based Knowledge Distillation"
_TMLR — Rejected by TMLR_

### Review · Reviewer_thpQ · 2025-11-08

**Summary Of Contributions:**

The papers main contribution is to challenge the assumption that chain‑of‑thought traces used for KD or SFT must be semantically correct and human‑interpretable to boost task performance. It tests this via a rule‑based problem decomposition that produces verifiably correct and incorrect traces for open‑book Question-Ans, while always training to the correct final answer Also runs a 100‑person user study on interpretability/cognitive load. Main empirical findings: (a) limited alignment between trace correctness and final answer correctness (b) a performance–interpretability disconnect/mismtach: training on verbose R1 traces often yields the best accuracy, but users rate those traces as least interpretable/most cognitively demanding .

**Audience:**

Yes

**Audience Explanation:**

this paper tackles an important and timely question: are the traces we train on the same traces we should show users? The combination of contributions being claimed in the paper -- verifiable trace construction, evaluation done on multiple datasets, and a realworld user study is valuable.

**Claims And Evidence:**

No

**Claims Explanation:**

I am worried that they key negative claim may largely reflect the training objective rather than an intrinsic property of reasoning traces. For the “SFT w/ Incorrect Traces” condition (Sec. 3.2 for setup, 4.2 for the experiment, sec 5 for results), every training sample pairs incorrect traces with the correct answer. Under the standard next‑token loss, the student may ignore the trace channel entirely and learn to output the answer from the input alone. This design sort of bakes in a decoupling within the experimental design itself between traces and answers and therefore strongly predisposes the model to produce “correct final answer + incorrect trace” at test time—which is precisely the phenomenon later interpreted as evidence of a lack of correlation. Is it possible to ensure that the student model specifically uses the trace for training? Say by adding an auxiliary/regularization loss so the model is forced to produce not jsut the output but also the intermediate trace ? Another way would be to ablate by giving partially correct traces.

Sec 4.3.1 — why do you keep only training cases where R1 answered correctly? That selection can shift the training distribution (potential data leakage toward “easy” items), no ? Could you report dataset sizes per condition and re‑run with identical counts sampled from the same item pool to account for such bias?

MARCO — why do you report accuracy rather than something like a ranking metric ? Is it possible that exact matching being tested for accuracy is under-estimating the actual performance?

For stronger validation, you could expand to more open-book Question Ans settings such as math-word problems which should give stronger indication of correctness or incorrectness of the trace.

User-study — is it possible that the users are confounding longer traces with cognitive load (rather than style, structure or other metrics of information content) ? It would make sense to control for length of traces (similar lengths) before accounting for their loads to make a stronger claim for correctness/usefulness. Also, each partio]ant only saw 5 examples which seem too few to make a reasonable conclusion. You could also report accuracy vs. token budget curves.

**Requested Changes:**

Please see concerns about the experimental setup above.

Missing related works:
Chain‑of‑Thought prompting elicits reasoning in LLMs (Wei et al., 2022) and Self‑Consistency (Wang et al., 2022) are important works for the claim that verbose traces can help accuracy.
Other ways to design experiments (beyond rule-based splits): Least‑to‑Most Prompting (Zhou et al., 2022), Program‑of‑Thoughts (Chen et al., 2023)

Minor: Some of the notations are inconsistent e.g. you use CoTemp but then Table 2 caption uses “Cotemporal”
There are some typos:
Page 2: tree -> three
Page 3 “and and”
Page 9: “Figure 3^2”
Page 11: “Acts provides support”
Broader impact: “the the”

---

> ### Author Response · Authors · 2025-11-26
> **Rebuttal**
>
> **W1 “key negative claim may largely reflect the training objective rather than an intrinsic property of reasoning traces”**: Our SFT setup ensures that the fine-tuned model generates a reasoning trace followed by a final solution/answer at the time of inference, similar to the training data. We agree with the reviewer’s hypothesis and explicitly highlight this through our work that the student model can learn to generate a correct final solution preceded by a semantically incorrect trace thereby implying that the semantics of the trace do not hold any importance in the LLM’s final performance. The student model has higher trace accuracies (as shown by trace evaluations in Table 2 and Table 3) when fine-tuned with <correct trace, correct solution> pairs than when fine-tuned with <incorrect trace, correct solution> pairs across all three datasets.
>
> **W2 “keep only training cases where R1 answered correctly”**: This experimental design decision allows us to create a uniform training dataset from prompting the R1. This is similar to how other distillation experiments are carried out where the distillation dataset for the smaller models can be generated by prompting the larger model unless the distillation is carried out using both positive and negative samples. In our case, we only consider the positive training samples. Hence, we get the summaries and post-hoc explanations for this filtered dataset. Ideally, we would have liked to create a dataset filtered on the basis of trace evaluations which is infeasible with R1.
>
> **W3 “report accuracy rather than something like a ranking metric”**: We would like to clarify that our choice of exact-match accuracy is directly aligned with the evaluation protocol established by the MS MARCO dataset creators. As documented in the original MS MARCO paper [1], the authors explicitly use exact matching (accuracy or precision–recall variants) for categories where answers are short. Since the subset of MS MARCO used in our experiments consists of questions with short, well-defined answers, our use of accuracy is consistent with the dataset’s intended evaluation scheme.
>
> **W4 “expand to more open-book Question Ans settings such as math-word problems”**: OpenBook QA domain is not only representative of a critical application of today's interactive dialogue systems, but more importantly offers a controlled experiment environment for demonstrating the disconnect between trace fidelity and final accuracy. In other reasoning areas, as shown by commonly used test datasets like AIME/MATH500 for math and LiveCodeBench for code, there are many problems where creating a step-by-step reasoning process that works for every math or code problem is not possible. This creates a problem for evaluating the intermediate steps in these situations. The main point is to demonstrate a gap between the accuracy of the steps and the final result for OpenBook QA problems. We have included this discussion in the paper on $\underline{page}$ $\underline{12}$, $\underline{Section}$ $\underline{5.5}$.
>
> **W5 “users are confounding longer traces with cognitive load”**: As expected, R1 traces are substantially longer and more verbose. However, our results indicate that trace length alone does not explain model performance as longer traces leading to better performance for the model is an orthogonal issue. R1-style traces still lead to significantly better task performance than their shorter counterparts, and have been rated as the lowest among all interpretability metrics answered by questions in Table 4. Note that these interpretability metrics including comprehensibility, faithfulness, etc. focus more on semantics than on just the user frustration which has been independently analyzed using the NASA TLX questions. This supports our core claim that end-user interpretability and LLM solution accuracy can be decoupled. Additionally, we have also added a trace-length analysis for all four conditions (R1, summarized R1, post-hoc explanations, and algorithmically correct traces) to the paper in $\underline{Appendix}$ $\underline{A.3}$, $\underline{page}$ $\underline{19}$.
>
> **RC1 “Additional related works” and RC2 “Notation consistencies”**: We thank the reviewer for pointing out these concerns, and we have now addressed each of these in the revised PDF.
>
> **References**:
>
> [1] Nguyen, Tri, Mir Rosenberg, Xia Song, Jianfeng Gao, Saurabh Tiwary, Rangan Majumder, and Li Deng. "Ms marco: A human-generated machine reading comprehension dataset." (2016).

---

### Review · Reviewer_Z9dj · 2025-11-10

**Summary Of Contributions:**

This paper investigates whether Chain-of-Thought (CoT) traces must be semantically correct and human-interpretable to improve LLM task performance when SFTed on those reasoning traces. The authors propose a rule-based problem decomposition on QA datasets to create verifiable correct/incorrect intermediate traces. They then conduct supervised fine-tuning (SFT) experiments and compare models trained with correct vs incorrect traces. Additionally, they perform a human subject study (n=100) evaluating interpretability of different trace types (R1, summarized R1, post-hoc explanations, algorithmically correct). The key findings are: (1) trace correctness doesn't reliably correlate with answer correctness; (2) most performant traces (R1) are least interpretable to humans.

### Strengths
- The paper challenges a common implicit assumption in current LLM training practices about trace validity
- The proposed rule-based decomposition enables verification of intermediate trace correctness
- Large scale human study with 100 participants and proper statistical analysis (Mann-Whitney U tests, Bonferroni correction)
- The paper provides a clear message on decoupling model supervision from user-facing explanations


### Weaknesses
- The authors only evaluate QA tasks leaving it unclear if findings generalize to other domains where CoTs are commonly used (coding, math, planning).
- The algorithmic "correct" traces are quite different in style from R1 traces, confounding length/verbosity with correctness. Ideally, the authors would provide statistics on trace lengths for each type (R1, summarized R1, post-hoc explanations, algorithmically correct). The R1 traces are presumably much longer, making them more frustrating and time-consuming for humans to read.
- The focus on training trace interpretability raises conceptual question: Should we care more about traces models produce at inference time? Training data should be useful for the model, not necessarily humans. Why not evaluate inference traces directly?

**Additional Comments:**

### Minor issues
- Citation format: Use \citep for citations in text for consistency
- Page 2, footnote 1: "tree parts" should be "three parts"
- Introduction: "asked to interpretability of the trace types using". You are missing a word in this sentence.
- Figure 1 caption could be more descriptive about what makes traces "verifiable"
- Some abbreviations used before definition (e.g., "SLM" on page 3)
- Reference formatting inconsistent (some have full author lists, others "et al.")

### Questions
- What's the reason for considering only 11 categories in Facebook bAbI QA dataset?
- On Microsoft MARCO QA and Facebook bAbI QA the trends are reversed compared to CoTemp. What's your hypothesis for why this is the case?
- For the incorrect reasoning traces, do you choose the incorrect problem type randomly? Or is there some systematic way of choosing it? In Table 1, the example of an incorrect reasoning trace given for Qwen3-1.7b has the same question type as the correct trace. How often does this happen?
- It could be interesting to investigate why incorrect traces sometimes help or what model-internal features they provide but this is probably beyond the scope of this paper. Do you have any preliminary thoughts on this?

**Audience:**

Yes

**Audience Explanation:**

The paper is relevant to current trends in the post-training literature and directly addresses an important question in current LLM training practices, i.e., whether intermediate traces need to be correct and human interpretable to be useful for model performance

**Broader Impact Concerns:**

The paper includes Broader Impact section discussing user trust and misinformation risks

**Claims And Evidence:**

Yes

**Claims Explanation:**

- The confusion matrices (Figures 2-3) clearly show the disconnect between trace/answer correctness across multiple models and datasets
- Authors perform proper hypothesis testing with appropriate corrections for human study

 **Minor concerns**:
  - Single fine-tuning run per condition; multiple runs would better account for variance

**Requested Changes:**

The following changes would strengthen the work:

- The definition of "interpretability" for CoT reasoning traces should be made more explicit. I'm not convinced that interpretability is the best term for what is being measured. Consider re-framing as "user-comprehensibility" of traces.
- Add a discussion on why findings might/might not transfer to other domains (mathematical reasoning, code generation)
- Investigate what properties of incorrect traces help models (length, token diversity, structural patterns)
- Create intermediate trace variants that control for length while varying correctness
- Run multiple fine-tuning iterations per condition to account for variance and ensure robustness
- Provide more trace examples: Point to examples in Appendix B.5 earlier in the paper and include more examples showing how different trace types differ in practice

---

> ### Author Response · Authors · 2025-11-26
> **Rebuttal**
>
> **W1 “findings generalize to other domains where CoTs are commonly used (coding, math, planning)”**: OpenBook QA domain is not only representative of a critical application of today's interactive dialogue systems, but more importantly offers a controlled experiment environment for demonstrating the disconnect between trace fidelity and final accuracy. In other reasoning areas, as shown by commonly used test datasets like AIME/MATH500 for math and LiveCodeBench for code, there are many problems where creating a step-by-step reasoning process that works for every math or code problem is not possible. This creates a problem for evaluating the intermediate steps in these situations. The main point of this work is to demonstrate a gap between the accuracy of the steps and the final result for OpenBook QA problems. We also note that there are other contemporaneous works that seem to show a similar disconnect between intermediate tokens and final solution accuracy in the context of planning-type scenarios [5]. We have included this discussion in the paper on $\underline{page}$ $\underline{12}$, $\underline{Section}$ $\underline{5.5}$.
>
> **W2 “statistics on trace lengths for each type”**:  In response, we have added a trace-length analysis for all four conditions (R1, summarized R1, post-hoc explanations, and algorithmically correct traces) to the paper (added in blue in $\underline{Appendix}$ $\underline{A.3}$, $\underline{page}$ $\underline{19}$). As expected, R1 traces are substantially longer and more verbose. However, our results indicate that trace length alone does not explain model performance: despite being longer and more cumbersome for humans to read, R1-style traces still lead to significantly better task performance than their shorter counterparts. This supports our core claim that end-user interpretability and LLM solution accuracy can be decoupled.
>
> **W3 “care more about traces models produce at inference time”**: We clarify that all our trace and final answer evaluations have been done at inference time. The results shown in Table 2 and Table 3 are from inference-time trace evaluations. We agree with the reviewer that “Training data should be useful for the model, not necessarily humans”, this is precisely our point too–which we have also mentioned both in the Abstract and the Broader Impact sections.
>
> **RC1 “definition of "interpretability"**: In this work, we intentionally focus on end-user interpretability, rather than mechanistic interpretability. Mechanistic interpretability focuses on internal computational mechanisms within the model, whereas end-user interpretability concerns how interpretable the model’s reasoning appears to human users. We have made this definition clear on $\underline{page}$ $\underline{7}$. The properties we measure (predictability, comprehensibility, interpretability, and faithfulness) are standard operationalizations of end-user interpretability used across the HRI communities [1,2,3,4]. Following this literature, we adopt these measures as validated proxies for end-user interpretability. To reflect this distinction more clearly, we have revised the paper to consistently use the term end-user interpretability throughout.
>
> **RC2 “discussion on why findings might/might not transfer to other domains”**: As discussed above for W1, we have included this discussion in the paper on $\underline{page}$ $\underline{12}$, $\underline{Section}$ $\underline{5.5}$.
>
> **References**:
>
> [1] Finale Doshi-Velez and Been Kim. Towards a rigorous science of interpretable machine learning, 2017.
>
> [2] Jalali, A., Haslhofer, B., Kriglstein, S., Rauber, A. (2023). Predictability and Comprehensibility in Post-Hoc XAI Methods: A User-Centered Analysis. In: Arai, K. (eds) Intelligent Computing. SAI 2023. Lecture Notes in Networks and Systems, vol 711. Springer, Cham. https://doi.org/10.1007/978-3-031-37717-4_46
>
> [3] Himabindu Lakkaraju and Osbert Bastani. 2020. "How Do I Fool You?": Manipulating User Trust via Misleading Black Box Explanations. In Proceedings of the AAAI/ACM Conference on AI, Ethics, and Society (New York, NY, USA) (AIES ’20). Association for Computing Machinery, New York, NY, USA, 79–85. https://doi.org/10.1145/3375627.3375833
>
> [4] Mahsan Nourani, Samia Kabir, Sina Mohseni, and Eric D Ragan. 2019. The effects of meaningful and meaningless explanations on trust and perceived
> system accuracy in intelligent systems. In Proceedings of the AAAI Conference on Human Computation and Crowdsourcing, Vol. 7. 97–105.
>
> [5] Stechly, Kaya, Karthik Valmeekam, Atharva Gundawar, Vardhan Palod, and Subbarao Kambhampati. "Beyond semantics: The unreasonable effectiveness of reasonless intermediate tokens." arXiv preprint arXiv:2505.1377

---

> > ### Author Response · Authors · 2025-11-26
> > **Rebuttal (continued)**
> >
> > **RC3 “what properties of incorrect traces help models”**: While there are ongoing contemporary efforts in understanding the workings of LLMs and how post-training techniques improve models’ performance on various benchmarks such as those on math and coding problems, it is hard to specifically pinpoint what is/are the causal reasons behind these models showing certain characteristics or changes in performance. In this work, we restricted our scope to specifically look at the correlation between semantic (in)correctness of traces with final solution performance. In the case of incorrect traces helping the model improve performance, we hypothesize that this boost in performance can possibly be attributed to the traces’ structural patterns seen by the model during training ignoring the semantic aspects. Both, correct and incorrect traces share the same structure in our controlled study setup, and the SFT learning regime that utilizes cross entropy loss possibly enforces the model to learn and output the <incorrect trace, correct final answer> at inference time.
> >
> > **RC4 “intermediate trace variants that control for length while varying correctness”**: Length of the traces and semantic correctness of the traces are two orthogonal tangents which can be varied in parallel to design several more experiment settings. In this work, we picked one specific setting for the length variant when constructing the two SFT training data setups (correct traces w/ correct final solutions & incorrect traces w/ correct final solutions) for all three datasets. We point out this assumption in our experiment design in $\underline{Section}$ $\underline{5.5}$ below the three key takeaways.
> >
> > **RC5 “multiple fine-tuning iterations per condition”**: We have additionally run the three SFT experiment settings - SFT Vanilla (with no traces), SFT w/ correct traces, and SFT w/ incorrect traces across three seeds and two models (Llama 3.1-1B and Qwen 3-1.7B) for the CoTemp QA dataset. The results are as follows and we see consistent results even across multiple seeds as presented in Table 2 in the paper.
> >
> > ---
> >
> > | Model         | Query Setting         | Final Solution Accuracy (Avg ± Std Dev) | Trace Acc - Classification Step (%) (Avg ± Std Dev) | Trace Acc - IR Step (%) (Avg ± Std Dev) | Trace Length (# tokens) (Avg ± Std Dev) |
> > |---------------|----------------------|--------------------------|-----------------------------------------------------|------------------------------------------|------------------------------------------|
> > | Qwen3-1.7b    | SFT - Vanilla        | 57.24 ± 1.25             | -                                                   | -                                        | -                                        |
> > | Qwen3-1.7b    | SFT - Correct Trace  | 49.08 ± 2.08             | 30.24 ± 1.76                                         | 47.73 ± 4.23                             | 28.38 ± 2.25                             |
> > | Qwen3-1.7b    | SFT - Incorrect Trace| 42.33 ± 2.66             | 22.40 ± 1.61                                         | 52.40 ± 5.54                             | 32.73 ± 2.83                             |
> > | Llama-3.2-1b  | SFT - Vanilla        | 43.73 ± 1.13             | -                                                   | -                                        | -                                        |
> > | Llama-3.2-1b  | SFT - Correct Trace  | 38.48 ± 0.10             | 39.66 ± 0.39                                         | 79.94 ± 0.30                             | 43.89 ± 0.20                             |
> > | Llama-3.2-1b  | SFT - Incorrect Trace| 36.55 ± 0.90             | 18.80 ± 0.57                                         | 75.62 ± 0.72                             | 40.62 ± 0.06                             |
> >
> > ---
> >
> > **RC6 “more trace examples”**: We have included multiple examples of the  R1 traces, summarized traces, post-hoc explanations, correct and incorrect traces  in $\underline{Appendix}$ $\underline{B.5}$, $\underline{pages}$ $\underline{25-32}$.

---

> > > ### Author Response · Authors · 2025-11-26
> > > **Rebuttal (continued)**
> > >
> > > **Q1 “reason for considering only 11 categories”**: We have selected 11 categories from the bAbI QA dataset which have representative numbers of train/test data samples in the original dataset. Due to the original dataset being significantly imbalanced, the other categories do not have sufficient train/test data samples and thus, were not utilized as part of the experimental setup.
> > >
> > > **Q2 “trends are reversed compared to CoTemp”**: As shown in the results for Table 2 and Table 3 for final solution performance, we see that SFT w/ incorrect traces outperforms SFT-Vanilla (no trace setting) and SFT w/ correct trace settings for CoTemp QA; SFT w/ correct traces does the best for Microsoft MARCO QA, while SFT-Vanilla outperformed the others in Facebook bAbI QA datasets - thereby resulting in all three possible cases strongly rejecting the hypothesis that semantic correctness of the traces does not guarantee improved final solution performance. This is further emphasized by the consistent results seen across the three datasets for their trace evaluations at inference time where expectedly, SFT w/ correct trace setup led to intermediate traces being more accurate than SFT w/ incorrect trace setup.
> > >
> > > To further strengthen our argument, we have also conducted a χ² statistical test with the null hypothesis: Trace accuracy and answer accuracy are independent. The test was done for both Qwen3-1.7B and Llama-3.2-1B-Instruct for the SFT w/ Incorrect trace setting. With a degree of freedom of 1, alpha=0.05, the critical value is 3.841 and we obtained χ² to be 0.34 and 2.93 (both < 3.841) for the two models respectively. Hence, the null hypothesis is correct. We have included these results in the revised PDF on $\underline{page}$ $\underline{9}$, $\underline{Section}$ $\underline{5.2}$.
> > >
> > > **Q3 “choose the incorrect problem type randomly”**: Yes, for avoiding any experimental bias, we selected a unique trace from another problem in the dataset at random and paired with the problem and its corresponding correct final solution to generate the training data. At inference time, the models learn to output such incorrect traces but with the correct final solution corresponding to the queried problem. Regarding the model learning to output the same question type at inference time, as shown by the example in Table 1, this varies for each dataset given the different number of problem categories. For the specific example in Table 1, the observation is merely coincidental because the incorrect trace paired with the queried problem could have also belonged to another problem in the dataset of the same category (and thus we see the same problem category generated by the model during inference), as these traces were paired at random for constructing the train dataset.
> > >
> > > **Q4 “investigate why incorrect traces sometimes help or what model-internal features”**: As discussed above for RC3, we have included our hypothesis on the interpretation of these results in the paper on $\underline{page}$ $\underline{8}$, $\underline{Section}$ $\underline{5.1}$.
> > >
> > > **Minor concerns**: We thank the reviewer for pointing out these concerns, and we have now addressed each of these in the revised PDF.

---

> > > > ### Comment · Reviewer_Z9dj · 2025-12-18
> > > > **Reply to rebuttal**
> > > >
> > > > Thanks for the detailed rebuttal and addressing and answering my concerns and questions. I will take this into consideration for my final recommendation.

---

### Review · Reviewer_b8oC · 2025-11-18

**Summary Of Contributions:**

The paper investigates the prevailing assumption in Chain-of-Thought (CoT) Knowledge Distillation (KD) that reasoning traces must be semantically correct and human-interpretable to be effective. The authors focus on the Question Answering (QA) domain, specifically Open Book QA, and employ a rule-based problem decomposition method to generate "verifiably correct" and "verifiably incorrect" traces.

The study involves two primary experimental thrusts:

 - Trace Correctness vs. Performance: Small Language Models (Llama-3.2-1B, Qwen3-1.7B) are fine-tuned (SFT) on datasets where traces are either correct or incorrect (containing wrong categories/facts), while the final answer remains correct. The authors find that models fine-tuned on incorrect traces often perform comparably to, or even better than, those trained on correct traces. Furthermore, they demonstrate that generating a correct trace is not reliably correlated with generating a correct final answer.

 - Interpretability vs. Performance: Larger models are fine-tuned on DeepSeek R1 traces, summaries of those traces, post-hoc explanations, and rule-based correct traces. A human-subject study (n=100) is conducted to rate these traces. The results show an inverse relationship: DeepSeek R1 traces yield the highest model performance but are rated lowest by humans for interpretability and highest for cognitive load. Conversely, the rule-based traces are deemed most interpretable but result in lower model accuracy.

**Audience:**

Yes

**Audience Explanation:**

The paper is about chain of thought reasoning for LLMs. This is a major area of current interest.

**Broader Impact Concerns:**

This paper highlights a potential risk in deploying reasoning models: the "illusion of explainability." If models can achieve high accuracy with unintelligible or even hallucinatory traces (as suggested by the R1 vs. Human ratings), users might be misled if they rely on those traces for verification. The work encourages the field to treat performance optimization and explainability as separate objectives, potentially requiring distinct modules, which is a valuable guiding principle for future AI safety and UX research.

**Claims And Evidence:**

Yes

**Claims Explanation:**

## Strengths

- Critical Examination of Core Assumptions: The paper tackles a timely and critical topic in the "Reasoning Model" era (e.g., DeepSeek R1, o1). The finding that the utility of a trace for a model is distinct from its interpretability to a human is a significant contribution that challenges the dual-purpose often assigned to CoT (performance booster + explanation).

- Novel Experimental Design (Incorrect Traces): The construction of the "SFT w/ Incorrect Traces" setting is a clever control. By forcing the model to train on logical non-sequiturs (wrong facts/categories leading to correct answers), the authors isolate the value of the content of the trace versus the mere presence of intermediate tokens.

- Rigorous Human Evaluation: Unlike many papers that rely on proxy metrics for interpretability (like length or perplexity), this work employs a well-structured human-subject study using established scales (Likert for faithfulness/comprehensibility and NASA-TLX for cognitive load). The sample size (100 participants) is robust for this type of ML study.

- Clear Verification Methodology: The use of rule-based decomposition for Open Book QA allows for objective ground-truth evaluation of intermediate steps (Classification Step and Information Retrieval Step). This avoids the ambiguity often found in evaluating free-form CoT reasoning.

## Weaknesses

- Domain Limitation (QA vs. Reasoning): The experiments are limited to Open Book QA tasks (CoTemp QA, MS MARCO, bAbI). While these require reasoning, they differ significantly from the complex mathematical or coding tasks where "Reasoning Models" like R1 typically shine. The rule-based decomposition used here is somewhat rigid; it is unclear if the finding (that incorrect traces don't hurt performance) would hold in math problems where a derivation error usually necessitates a wrong answer.

- Nature of "Incorrect" Traces: The "incorrect" traces are constructed by swapping in wrong problem categories or facts. It is possible that the models in the "Incorrect Trace" setting are simply treating the trace as noise/padding and performing standard input-output mapping (effectively reverting to standard SFT without CoT, but with extra tokens). The paper briefly mentions this ("albatross"), but a deeper analysis of why incorrect traces work (e.g., attention analysis) is missing.

- Generalization of "R1 Traces": The paper concludes R1 traces are uninterpretable based on QA tasks. R1 traces are often exploratory (backtracking, self-correction). In QA, this verbosity might look like hallucination or rambling, whereas in Math, it looks like genuine problem solving. The context dependence of "interpretability" could be discussed more.

**Requested Changes:**

To strengthen the submission for TMLR, I recommend the authors address the following:

- Baseline Comparison: In the "SFT w/ Incorrect Traces" experiments, how does the performance compare to a model trained without any traces (standard Input-Output SFT)? Table 2 includes "SFT Vanilla", and "SFT w/ Incorrect Traces" often beats it. Please add a brief discussion hypothesizing why nonsense traces help more than no traces. Is it simply increased computation time (more tokens to process context)?

- Qualitative Analysis of Incorrect Traces: Provide examples of the "Incorrect Traces" generated by the model at inference time. Does the model learn to generate the specific incorrect patterns shown in training, or does it generate its own hallucinations? This would clarify if the model learned the "style" of incorrectness.

- Scope Disclaimer: Explicitly acknowledge the gap between Open Book QA (retrieval-heavy) and Logic/Math (derivation-heavy). The finding that "trace correctness is not correlated with answer correctness" might be specific to tasks where the answer can be retrieved directly from context, rendering the trace redundant. Acknowledging this boundary condition would make the claims more precise.

- Clarify "Semantic Correctness": In Section 1, clarify that "validity" and "semantic correctness" in this context refer specifically to the alignment with the rule-based decomposition (Category and IR steps).

---

> ### Author Response · Authors · 2025-11-26
> **Rebuttal**
>
> **W1 “Domain Limitation (QA vs. Reasoning)”**: OpenBook QA domain is not only representative of a critical application of today's interactive dialogue systems, but more importantly offers a controlled experiment environment for demonstrating the disconnect between trace fidelity and final accuracy. In other reasoning areas, as shown by commonly used test datasets like AIME/MATH500 for math and LiveCodeBench for code, there are many problems where creating a step-by-step reasoning process that works for every math or code problem is not possible. This creates a problem for evaluating the intermediate steps in these situations. The main point is to demonstrate a gap between the accuracy of the steps and the final result for OpenBook QA problems. As discussed above, we have included this discussion in the paper on $\underline{page}$ $\underline{12}$, $\underline{Section}$ $\underline{5.5}$.
>
> **W2 “Nature of "Incorrect" Traces”**: While there are ongoing contemporary efforts in understanding the workings of LLMs and how post-training techniques improve models’ performance on various benchmarks <cite>, it is hard to specifically pinpoint what is/are the causal reasons behind these models showing certain characteristics or changes in performance. In this work, we restricted our scope to specifically look at the correlation between semantic (in)correctness of traces with final solution performance. In the case of incorrect traces helping the model improve performance, we hypothesize that this boost in performance can possibly be attributed to the traces’ structural patterns seen by the model during training ignoring the semantic aspects. Both, correct and incorrect traces share the same structure in our controlled study setup, and the SFT learning regime that utilizes CE loss possibly enforces the model to learn and output the <incorrect trace, correct final answer> at inference time. We have included this discussion in the paper on $\underline{page}$ $\underline{8}$, $\underline{Section}$ $\underline{1}$.
>
> **W3 “Generalization of "R1 Traces”**: Our analysis focuses on end-user interpretability, not mechanistic interpretability. While mechanistic interpretability concerns a model’s internal computations, end-user interpretability captures how understandable the model’s reasoning appears to human users, as defined in the paper ($\underline{page}$ $\underline{7}$). Therefore, all conclusions drawn in the paper regarding interpretability are with context to end-user interpretability. We have revised the paper to make this clear.
>
> **RC1 “Baseline Comparison”**: In both Table 2 and Table 3, we report the final solution accuracy for the standard input-output SFT experiment and see varying results across the three domains. As shown in the results for Table 2 and Table 3 for final solution performance, we see that SFT w/ incorrect traces outperforms SFT-Vanilla (no trace setting) and SFT w/ correct trace settings for CoTemp QA; SFT w/ correct traces does the best for Microsoft MARCO QA, while SFT-Vanilla outperformed the others in Facebook bAbI QA datasets - thereby resulting in all three possible cases strongly rejecting the hypothesis that semantic correctness of the traces does not guarantee improved final solution performance. As discussed in our response above to W2, we have included this hypothesis/discussion in the paper on $\underline{page}$ $\underline{9}$, $\underline{Section}$ $\underline{5.2}$.
>
> **RC2 “Qualitative Analysis of Incorrect Traces”**: We have provided multiple examples of the incorrect traces  in $\underline{Appendix}$ $\underline{B.5}$, $\underline{pages}$ $\underline{25-32}$.  Based on these examples, we observe that the model tends to learn the style or pattern of “incorrectness” present in the training data, rather than producing new hallucinations.
>
> **RC3 “Scope Disclaimer”**: We have included this discussion on crisply defining the scope of this work in the paper on $\underline{page}$ $\underline{12}$, $\underline{Section}$ $\underline{5.5}$.
>
> **RC4 Clarify "Semantic Correctness"**: We thank the reviewer for pointing out this concern, and we have now addressed this in the revised PDF on $\underline{page}$ $\underline{2}$, $\underline{Section}$ $\underline{1}$.

---

### Author Response · Authors · 2025-11-26
**Rebuttal Overview**

We thank all the reviewers for the time and feedback, and have addressed each of the questions/concerns! In response to these comments (some of them commonly asked by more than one reviewer), we have revised the PDF (changes highlighted in $\textcolor{blue}{blue}$ text) to include the suggested clarifications in the paper.

In summary,

1. **(Reviewer Z9dj, thpQ, b8oC) Precisely defining the scope of the work**: We clarify the scope of the work to be restricted to the Open Book QA settings and what challenges will be encountered when performing a similar study with other reasoning domains such as coding, math, etc. We have included this discussion in the paper on $\underline{page}$ $\underline{12}$ $\underline{in}$ $\underline{Section}$ $\underline{5.5}$.

2. **(Reviewer Z9dj, thpQ, b8oC) Trace-length analysis and qualitative analysis**: As part of our discussion on the trade-offs between LLM performance and end-user interpretability when using different trace types, we have included a trace-length analysis comparing the four different trace types used in our study, i.e., DeepSeek R1 traces, summarized R1 traces, post-hoc explanations of R1 traces, and algorithmically generated traces. This analysis is included in the paper in blue in $\underline{Appendix}$ $\underline{A.3}$, $\underline{page}$ $\underline{19}$. For the qualitative analysis, we have now included multiple examples of the  R1 traces, summarized traces, post-hoc explanations, correct and incorrect traces in $\underline{Appendix}$ $\underline{B.5}$, $\underline{pages}$ $\underline{25-32}$.

3. **(Reviewer Z9dj, b8oC) Hypothesis behind incorrect traces helping performance**: As we see in our SFT results in Table 2 and Table 3, the LLMs finetuned with incorrect traces but correct final solutions also improved final solution accuracy. For completeness of the study, we have included a discussion on the possible hypothesis why these traces could possibly lead to improved accuracy. We have included our hypothesis on the interpretation of these results in the paper on $\underline{page}$ $\underline{8}$, $\underline{Section}$ $\underline{5.1}$.

4. **(Reviewer Z9dj) Statistical testing for checking (lack of) correlation**: We have also conducted a χ² statistical test with the null hypothesis: Trace accuracy and answer accuracy are independent. We have included these results in the revised PDF on $\underline{page}$ $\underline{9}$, $\underline{Section}$ $\underline{5.2}$.

---

### Decision · Action_Editor_jAzH · 2026-01-01

**Recommendation:** Reject

**Audience:**

Yes

**Audience Explanation:**

This paper would be of interest to researchers working on SFT and related topics.

**Claims And Evidence:**

No

**Claims Explanation:**

This paper was reviewed by three expert reviewers and I am grateful for their feedback. All reviewers acknowledged that the paper addresses a timely problem and were impressed by the work that went into the human study. However, after the author response, the reviewers still had concerns that some of the claims made in the paper are not “supported by accurate, convincing and clear evidence.”

The reviewers’ main concern is that there are several places in the paper where the claims suggest a strong causal interpretation, but the experimental setup is not sufficient to justify these causal claims. For example, the abstract explicitly claims causal isolation: “to isolate the effect of trace semantics…” This framing implies that changing trace correctness isolates a causal factor. However, based on the experimental setup (incorrect trace + correct answer SFT), the training objective can be optimized by learning a direct map from input to answer, with the trace tokens treated as a kind of stylistic prefix. This suggests the decoupling that the paper observes between trace and answer correctness could be more a function of the specific training objective/design rather than a global intrinsic property of trace semantics. The Discussion also makes strong causal claims, such as “we were able to demonstrate that the correctness of training traces does not even matter in the first place.”

In the end, we recommend a reject with the option for a major revision, since the reviewers all agreed that the paper contains many interesting observations, but the claims and conclusions are too strong as written, and this is the primary criterion for TMLR. The authors are recommended to either carefully revise the paper to make sure the paper doesn't overclaim about causality (i.e., narrow the claims to explicitly describe what's shown under this particular SFT objective), or add experiments that better support the causality claims.

**Resubmission Of Major Revision:**

The authors may consider submitting a major revision at a later time.